# Controls on the southwest USA hydroclimate over the last six glacial-interglacial cycles

Kathleen A. Wendt [1,2,3] ✉, Stacy Carolin [4,5], Christo Buizert [2], Simon D. Steidle [1], R. Lawrence Edwards [6], Gina E. Moseley [1], Yuri Dublyansky [1], Hai Cheng [7,8], Chengfei He [9], Mellissa S. C. Warner [6] & Christoph Spötl [1]

The Great Basin in the southwest United States experienced major hydroclimate shifts throughout the Quaternary. Understanding the drivers behind these past changes has become increasingly important for improving future climate projections. Here, we present an absolute-dated $\delta^{18}O$ and $\delta^{13}C$ record from Devils Hole cave 2 (southern Nevada) that reveals climate and environmental changes in the southern Great Basin over the last 580,000 years. Water isotope-enabled Earth system simulations and phasing analysis show that temperature-related mechanisms are a primary driver of $\delta^{18}O$ variability, with additional drivers stemming from processes linked to North American ice sheets. Vegetation density in the highlands of southern Nevada is primarily forced by Northern Hemisphere summer intensity. A rapid decline in primary productivity occurs during warm interglacial periods when local groundwater recharge declines to <50% above modern. Our study sheds new light on the relationship between temperature, moisture balance, and vegetation over the last six glacial-interglacial cycles.

Throughout the Pleistocene, Earth's climate system oscillated between glacial and interglacial modes. These oscillations were paced by changes in Earth's orbital parameters, which alter the distribution of incoming solar radiation and, in turn, drive the advance and retreat of Northern Hemisphere ice sheets (i.e., Milankovitch theory). One of the most striking regional expressions of these climate modes is the repeated expansion of large pluvial lakes in the drylands of the southwest (SW) United States during glacial periods[1–6]. Reconstructing past hydro-climate variability on orbital timescales ($10^4$–$10^5$ years) remains a challenge due to limited dating precision and the discontinuous nature of terrestrial climate archives in the Great Basin. Devils Hole cave in southern Nevada provides a rare exception; the cave contains continuous calcite deposits that can be precisely dated

over multiple glacial-interglacial cycles. The first $\delta^{18}O$ timeseries from Devils Hole[7,8] sparked widespread debate over the timing of the penultimate deglaciation, also known as Termination (T) II. The Devils Hole record suggested that the transition to interglacial climate conditions in the Great Basin occurred ~10,000 years prior to the increase in Northern Hemisphere summer insolation[7,8]. This finding contradicted the widely supported Milankovitch theory, which attributes the pacing of glacial-interglacial transitions to changes in Earth's orbital configuration[9,10].

Recent work identified an offset in uranium (U)-series ages relative to water depth in Devils Hole (DH) cave and its adjacent cave, Devils Hole 2 (DH2)[11]. The offset was attributed to elevated concentrations of hydrogenous $^{230}Th$ with increasing depth, which may

[1]Institute of Geology, University of Innsbruck, Innrain 52, Innsbruck, Austria. [2]College of Earth, Ocean, and Atmospheric Sciences, Oregon State University, 101 SW 26th St., Corvallis, Oregon, USA. [3]Department of Earth Sciences, University of Toronto, Toronto, ON, Canada. [4]Department of Earth Sciences, University of Cambridge, Downing St, Cambridge, UK. [5]Department of Earth Sciences, University of Oxford, Oxford, UK. [6]Department of Earth Sciences, University of Minnesota, Minneapolis, Minnesota, USA. [7]Institute of Global Environmental Change, Xi'an Jiaotong University, Xi'an, China. [8]State Key Laboratory of Loess and Quaternary Geology, Institute of Earth Environment, Chinese Academy of Sciences, Xi'an, China. [9]Rosenstiel School of Marine, Atmospheric, and Earth Science, University of Miami, Coral Gables, FL, USA. ✉e-mail: kathleen.wendt@utoronto.ca

have biased the ages of subaqueous calcite deposited at significant (tens of meters) depth by tens of thousands of years[11]. In contrast, shallow subaqueous calcite deposits from DH2 suggested that the rise in interglacial δ[18]O values broadly coincided with the rise in Northern Hemisphere summer insolation associated with Termination II[11]. Importantly, the DH2 δ[18]O record agrees with other Nevada speleothem δ[18]O records from vadose caves[12–15], both in terms of the timing of Termination II and the duration of the last interglacial[11], thus reconciling the discrepancy between the DH record and other well-dated speleothem records in this region.

The DH2 record joins a growing body of evidence suggesting that the timing of Great Basin climate variability is consistent with the Milankovitch theory, yet the drivers of Great Basin δ[18]O change on orbital timescales remain poorly understood. The repeated expansion and desiccation of pluvial lakes in the Great Basin are a direct proxy for changes in the regional water balance. DH2 δ[18]O, however, is uncoupled from southern Great Basin palaeo lake and groundwater table records at various points in time (e.g., 2), suggesting that different or multiple controls of regional δ[18]O are at play. Furthermore, few δ[18]O records in this region span more than one glacial cycle. Identifying the controls of δ[18]O variability over glacial cycles is a critical step in disentangling the mechanisms that govern (hydro-)climate change in the Great Basin on orbital timescales. Understanding these mechanisms has become increasingly urgent, as warmer temperatures over the next century are expected to reduce water availability in this already water-scarce yet increasingly populated region[16–20].

Here, we extend the DH2 δ[18]O timeseries of ref. 11 from 204 to 508 thousand years before present (ka). The extended record captures Great Basin climate variations over six glacial-interglacial cycles with excellent chronological precision (2σ age uncertainties 0.3–2%). We use an isotope-enabled version of the Community Earth System Model (iCESM) to further investigate the controls of glacial-interglacial δ[18]O changes at DH2 cave. Lastly, we examine orbital-

scale environmental changes in southwest Nevada over the extended DH2 δ[13]C record.

## Results and discussion

### Devils Hole vs Devils Hole 2

The DH and DH2 caves are a set of extensional fractures located 200 m apart in a detached section of Death Valley National Park in southern Nevada (Supplementary Fig. S2). Both caves intersect a large (12,000 km²) slightly thermal (34 °C) groundwater flow system that is recharged from the melt of snowpack in the high elevations (> 2700 m a.s.l.) of the Spring and Sheep mountain ranges ~80 km to the south-east, with minor inputs from the high ranges of central Nevada ~400 km northeast (Supplementary Fig. S2)[21,22]. Groundwater transit times from recharge centers to DH and DH2 caves are estimated from hydrogeologic data to be 880 years[23,24] due to the groundwater system's low hydraulic gradient[22]. Groundwater transit times were likely shorter during glacial periods due to significantly higher groundwater recharge (up to +250%)[25]. The subvertical walls of DH and DH2 are coated with a thick (~1 meter) calcite crust that precipitated subaqueously from groundwaters (modern calcite saturation index = 0.2)[26].

The extended DH2 record was constructed using a shallow (i.e., near-surface) core (core D) from ref. 11 that has not been significantly affected by excess ²³⁰Th. Its chronology is anchored by 114 high-precision U-series ages (Supplementary Tables S1 and S2) where average 2σ dating uncertainties are 0.5% from 4.8–200 ka, 1% from 200–400 ka, and 2% from 400–736 ka (Fig. 1). Four growth hiatuses identified in core D coincide with peak interglacial periods associated with Marine Isotope Stages (MIS) 5e, 7a, 7e, and 9e (Supplementary Fig. S1), during which the water table in DH 2 fell below the elevation of core D (+1.8 m relative to modern water table [r.m.w.t.])[11,27,28]. Corresponding δ[18]O and δ[13]C data from a calcite core (core P) that grew continuously during interglacial periods and was retrieved at a lower

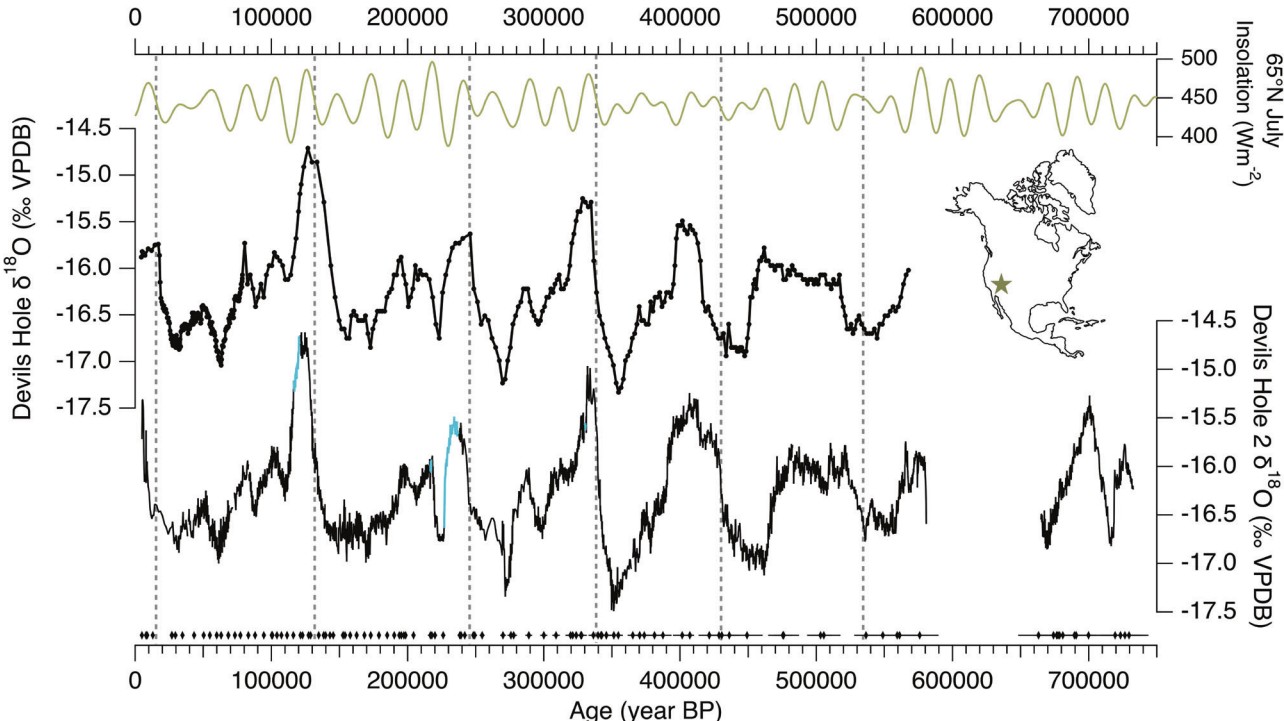

**Fig. 1 | Devils Hole (DH) versus Devils Hole 2 (DH2) records.** From top: 65°N July insolation[94], DH δ[18]O[7,30], and DH2 δ[18]O (11; this study). Blue lines in DH2 record indicate core-P δ[18]O data spliced into core D record (see text). DH2 U-series ages (black diamonds) with respective 2σ age uncertainties (horizontal black bars) from refs. 11,29,35 and this study (see Supplementary Table S1). Dashed vertical bars demarcate the rise in 65°N July insolation associated with Terminations I-VI. Location of DH and DH2 caves indicated by a star on the map.

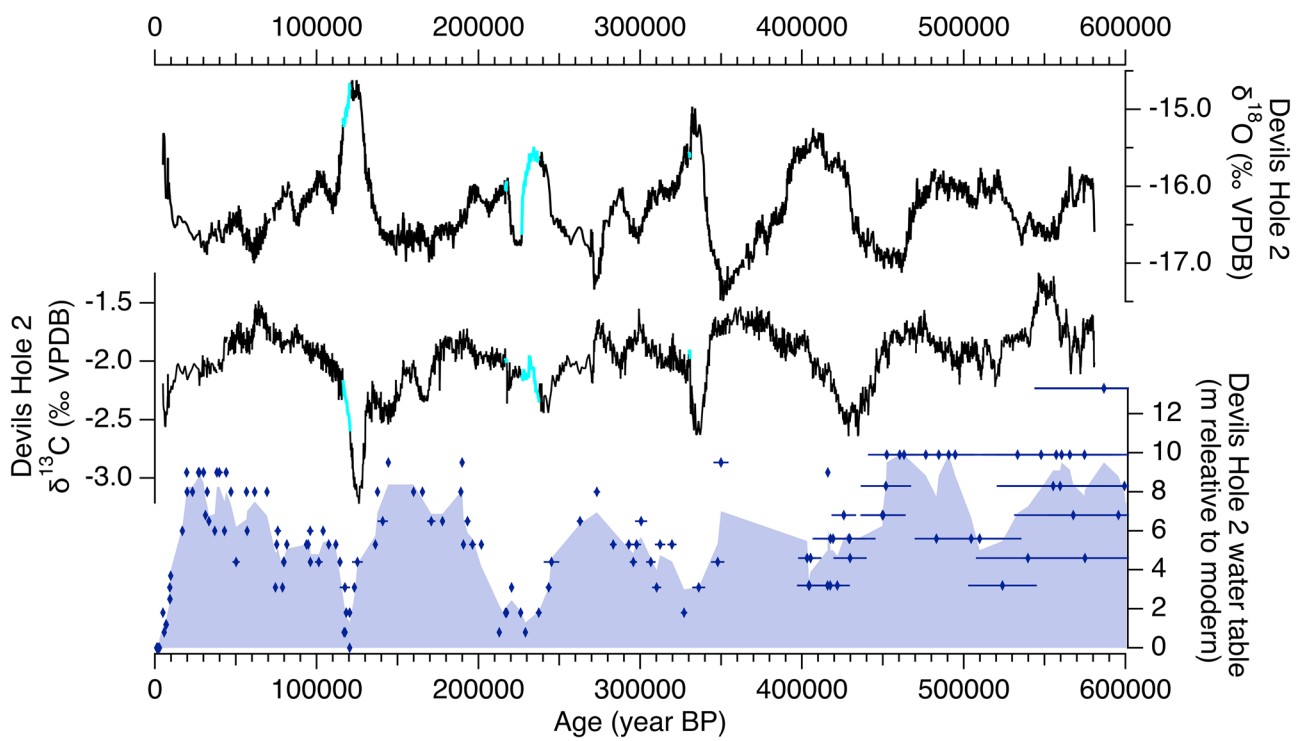

**Fig. 2 | Devils Hole 2 (DH2) $\delta^{18}O$, $\delta^{13}C$, and palaeo water table elevation over the last 600 ka.** From top: DH2 $\delta^{18}O$ and $\delta^{13}C$ (11; this study). Blue lines indicate core-P $\delta^{18}O$ data spliced into core D record (see text). Blue diamonds indicate palaeo water table elevations in DH2 cave relative to modern, including 2σ age uncertainties[27,28]. Purple shading is produced from a binomial spline of DH2 water table data.

(but still shallow) elevation (− 1.6 m r.m.w.t.) was spliced into the gaps of the core D record (see methods). A growth hiatus between 580–665 ka[29] has no known corresponding growth phases in lower elevation cores, and is also in agreement with the original work in DH[7], indicating that it is a widespread and not local cave feature. The mechanism(s) that drove the cessation or erosion of these subaqueous calcite deposits remain unknown.

The extended DH2 record can be compared to the original DH record[7] in terms of both its $\delta^{18}O$ features and chronology. The DH2 record confirms a majority of the $\delta^{18}O$ features of the original DH studies at a higher resolution (Fig. 1), demonstrating these to be a robust and reproducible climate record. Consistent with the original DH record, the magnitude of interglacial $\delta^{18}O$ maxima is lower prior to the mid-Bruhnes transition (430 ka), with the exception of MIS 17. As for the chronology, large portions of DH2 replicate the original DH chronology with the exception of TI-III, during which the higher-elevation DH2 record is several thousands of years younger than the original DH record, which was collected at significant depth below the water table (Supplementary Fig. S3). The shift to interglacial DH2 $\delta^{18}O$ values during TI-VI and VIII broadly coincides with the rise in boreal summer insolation associated with each T, and is thus consistent with mechanisms ultimately tied to orbital forcing[11].

A longstanding issue with the original DH record is in the timing of TII and the duration of the last interglacial (LIG). Previous studies argued that an "early" TII and TI recorded in DH $\delta^{18}O$ may be due early warming of sea surface temperatures (SSTs) along the California Current[30,31] yet other well-dated dated palaeo records in the Great Basin[3,12–15] and California[32,33] do not indicate an early onset of inter-glacial conditions. The DH2 record, by contrast, reconciles these regional discrepancies[11]. The timing of TII in DH2 agrees with other well-dated records in the southern Great Basin that reflect the $\delta^{18}O$ and $\delta^2H$ of palaeo rainfall[3,12–15] (Supplementary Fig. S11), albeit with different magnitudes of $\delta^{18}O$ variations due to differences in speleothem

type, deposition rate, and cave settings (Supplementary Text S1 and Supplementary Fig. S11). DH2 $\delta^{18}O$ suggests a LIG duration of ~6 kyrs, in agreement with Nevada stalagmite $\delta^{18}O$ and Mojave Desert $C_{31}$ n-alkane $\delta^2H$ records (duration of 4-6 kyrs)[3,13], contrasting with the DH $\delta^{18}O$ LIG duration of ~12 kyrs[7]. The DH2 record also shows better agreement with changes in local effective moisture. Palaeo water table reconstructions from DH[34] and DH2[27,28] caves reflect past changes in groundwater recharge amount to the local aquifer over the last 750 ka[25,27] (Fig. 2). At TII, the rise in DH2 $\delta^{18}O$ towards interglacial values coincides with a multi-meter drop in the local palaeo water table[11,27] and the desiccation of pluvial lakes in Death Valley and Searles Valley in the southern Great Basin (onset of drying at 137.6 ka ± 0.5 ka[2];), which occurred ~ 10 kyrs after the TII rise in DH $\delta^{18}O$[7]. The extended DH2 $\delta^{18}O$ record shows similar agreement in the timing of a multi-meter drop in the palaeo water table (e.g., local drying) associated with TIII-VI[27,28,34] (Fig. 2). The magnitude of DH2 $\delta^{18}O$ is also negatively correlated with DH2 $\delta^{234}U_i$ maxima during each glacial period[35], which is interpreted as a proxy for water-rock interactions associated with a fluctuating water table, with high values corresponding to periods of high groundwater recharge[35]. For example, exceptionally low $\delta^{18}O$ values and relatively high $\delta^{234}U_i$ maxima (1850‰) during Marine Isotope Stage (MIS) 10 suggest that this glacial period was exceptionally wet and cool in the southern Great Basin.

## Controls on orbital-scale $\delta^{18}O$ variations at Devils Hole 2

Today, over 90% of groundwater recharge to the DH2 aquifer originates from snowmelt in high elevation mountain ranges[21]. Snow accumulates during winter months when extratropical cyclones from the North Pacific move along the Pacific Storm Track (Fig. 3)[21,36]. In contrast, summer precipitation from the North American Monsoon (NAM) contributes < 10% of annual groundwater recharge[21]. Back trajectories of modern rainfall (Fig. 3) show that a majority of winter rainfall originates from 30-45°N and 135–150°W[37], whereas summer

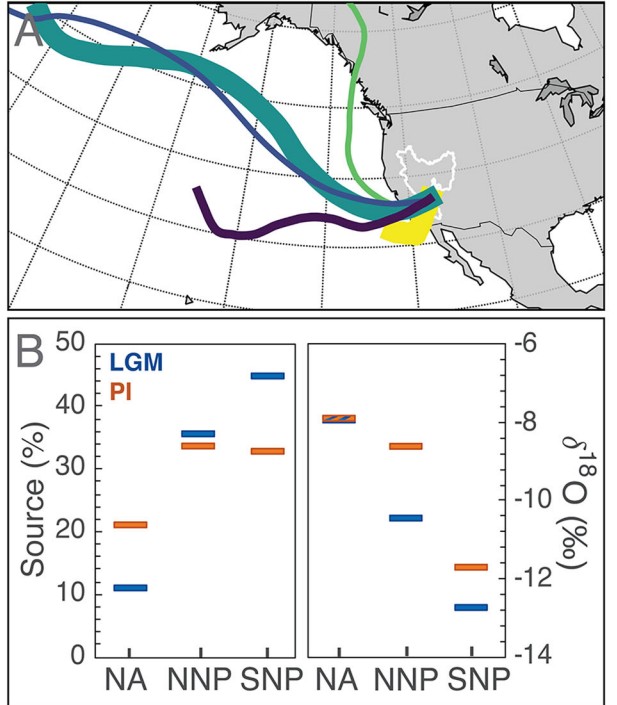

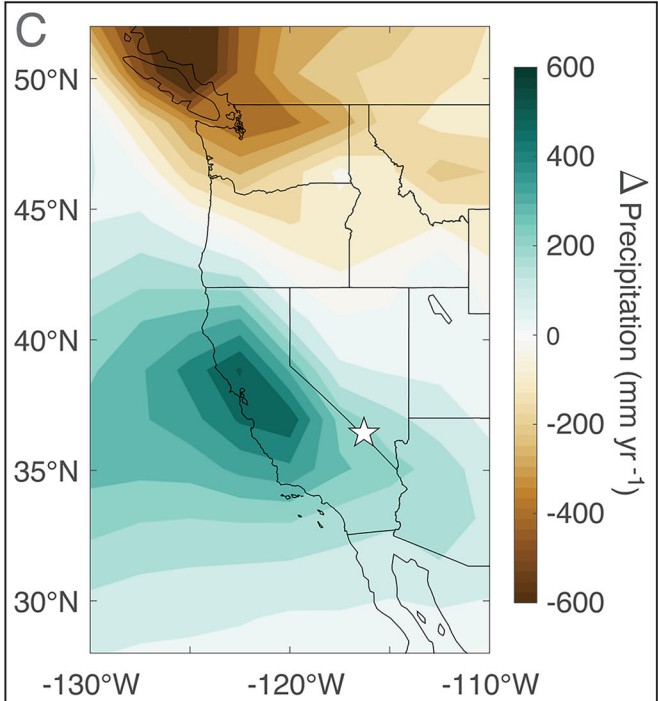

**Fig. 3 | Modern moisture trajectories and iCESM results. A** Cluster analyses of modern rain-bearing trajectories arriving at Devils Hole caves from September 2007 to August 2011, adapted from ref. 37. Great Basin is outlined in white. Colors are used to distinguish clusters, with the width of each cluster representative of the number of trajectories. **B** Modeled Last Glacial Maximum (LGM; blue) versus pre-industrial (PI; orange) change in source % and $\delta^{18}O$ of precipitation at Devils Hole caves sourced from the North American continent (NA), Northern North Pacific (NNP), and Southern North Pacific (SNP). All other minor moisture sources (< 5% source in PI) shown in Fig. S5. **C** Modeled LGM versus PI change in precipitation over western North America. Devils Hole caves indicated by a white star.

NAM rainfall originates from along the east coast of California and Baja Peninsula[38] and contributes ~10% of rain-bearing trajectories to DH2[37]. The southern Great Basin was significantly wetter during glacial periods, as spectacularly demonstrated by large pluvial lakes that expanded throughout this desert region. Wetter glacial conditions are attributed to cooler temperatures and suppressed evaporation[39,40] coupled with a southward displacement of the Pacific Storm Track[39,41–43] that resulted in a drier northwest United States and wetter southwest (opposite to modern day). Enhanced Pacific-sourced atmospheric rivers may have also contributed to increased moisture transport[39,44], whereas NAM is estimated to have weakened[38] during glacial periods.

The exact mechanisms that drive DH2 $\delta^{18}O$ depletion during glacial periods remain unclear. Previous studies[7,11] suggest that glacial-interglacial DH2 $\delta^{18}O$ variations are partially driven by changes in the proportion of summer precipitation sourced from NAM (high $\delta^{18}O$) to winter precipitation sourced from the Pacific (low $\delta^{18}O$) at DH2. As a result, low $\delta^{18}O$ values during glacial periods are partially due to a larger proportion of low-$\delta^{18}O$ cool-season rainfall associated with (i) a southerly displaced Pacific Storm Track and (ii) a weakened NAM. Another proposed mechanism is temperature: terrestrial proxies suggest that the southern Great Basin was 6–10 °C cooler during the last glacial maximum (LGM) relative to preindustrial[3,39,45,46] and SSTs reconstructed from the moisture source regions of DH2 were 2–5 °C cooler[47,48]. Tabor et al.[39] suggest that a greater land-sea temperature gradient during the LGM increased the rainout efficiency of moisture trajectories moving inland to the Great Basin. This effect, when coupled with cooler LGM temperatures that suppressed evaporation, would result in lower $\delta^{18}O$ of precipitation during glacial periods[39,49]. Groundwater temperatures in the DH2 aquifer have remained constant (± 1 °C) over the last 500 ka[50,51] and thus have a negligible effect on $\delta^{18}O$ variations in DH2 calcite.

To further investigate potential mechanisms, we used a water isotope-enabled Earth System Model (iCESM1.3) with moisture tagging to examine changes in $\delta^{18}O$ of precipitation at DH2[52,53]. LGM simulations show a 1.3‰ decrease in the annual average $\delta^{18}O$ of precipitation at DH2 relative to preindustrial (PI; 1850 CE) (Fig. 3). A depletion of water vapor $\delta^{18}O$ for all months is partially attributed to cooler LGM temperatures at DH2's moisture source regions. LGM simulations show a 50% increase in annual precipitation amount (Fig. 3). Increased precipitation occurred during winter months (Fig. 3 and Supplementary Fig. S4) during which there was enhanced transport of Pacific-sourced moisture. This is attributed to a southerly displaced Pacific Storm Track, as supported by the LGM-PI differences in the mean eddy kinetic energy (Supplementary Fig. S6b) and an increase in the vapor fraction sourced from the southern North Pacific (10°N to 30°N) relative to northern North Pacific (30°N to 60°N) (Supplementary Fig. S4). Despite its lower latitude source, southern North Pacific water vapor arrives to DH2 ~ 2‰ more depleted relative to northern North Pacific vapor (Fig. 3), likely due to longer moisture trajectory pathways and/or higher rainout efficiency resulting from a greater land-sea temperature gradient. The proportion of precipitation sourced from the North American continent decreased during the LGM (Fig. 3), likely due to suppressed re-evaporation from land sources as a result of cooler terrestrial surface temperatures. This iCESM insight agrees with a recent DH2 $^{17}O_{excess}$ study that suggests reduced continental recycling during glacials[54]. Because land-sourced vapor is relatively enriched in $\delta^{18}O$, a decrease in its contribution to DH2 precipitation during the LGM results in an overall depletion (Fig. 3). Finally, model results do not support a correlation between $\delta^{18}O$ change and precipitation amount. This finding agrees with proxy data, which show a decoupling of DH2 $\delta^{18}O$ from local effective moisture at various points in time. For example, MIS 5e DH2 $\delta^{18}O$ reaches maximum interglacial values at approximately 127 ka before plateauing for ~ 6 kyrs[3,13,14], whereas the

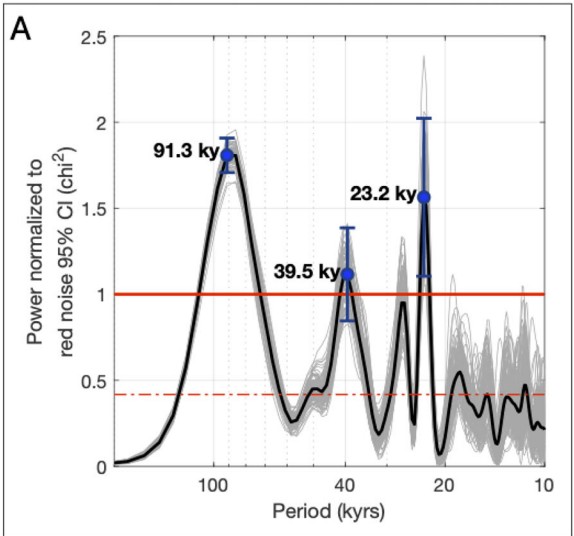

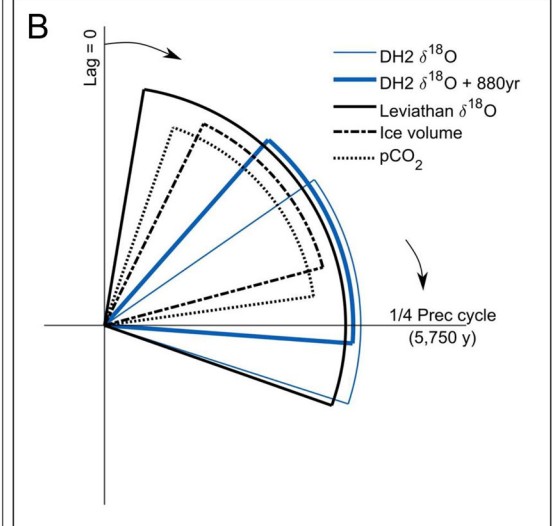

**Fig. 4 | Devils Hole 2 (DH2) $\delta^{18}O$ power spectrum and phasing analysis. A** DH2 $\delta^{18}O$ power spectrum of 100 OxCal age models. Each power spectrum has been normalized to its corresponding 95% false alarm record, such that power >1 is interpreted as a frequency significant above red noise. The black line is the mean normalized power of the 100 records calculated at each frequency. The blue circles highlight local maximum mean normalized power above 1, with 2σ error bars from the 100 records. The red dash dot line represents the theoretical red noise (first-order autoregressive process). **B** DH2 $\delta^{18}O$, Leviathan $\delta^{18}O$, global ice volume, and global atmospheric $CO_2$ (p$CO_2$) records' phase lag (95% CI) relative to the orbital precession index, in the 22–24 kyr period window. Zero phase (pointing up) is set as precession index minimum, equivalent to the Northern Hemisphere summer solstice (June 21st) insolation maximum, and arrows mark direction of increasing years of lag from the set zero phase (1 cycle = 23 kyr). Wedges are staggered in height for easier viewing. Datasets: DH2 $\delta^{18}O$ on its mean age model (this study) and adjusted 880 years to account for maximum groundwater recharge time (this study, blue), Leviathan $\delta^{18}O$ from Nevada stalagmites[14], atmospheric p$CO_2$ on the AICC2023 chronology[59,60] and global ice volume inferred from the absolute-dated Red Sea relative sea level (RSL)[58].

DH2 water table continues dropping until 120.3 ± 0.5 ka. A similar decoupling of DH2 $\delta^{18}O$ and local effective moisture is observed during interglacials MIS 7e, 7c, 9, and 11, within dating uncertainties.

In total, iCESM moisture tagging experiments suggest two key drivers of DH2 $\delta^{18}O$ variability on glacial-interglacial timescales. First, vapor delivered to DH2 during the LGM is more strongly depleted in $\delta^{18}O$ for all months due to cooler temperatures and temperate-driven rainout effects (Supplementary Fig. S4). Second, a change in the proportion of moisture from distinct sources, specifically (i) an increase in depleted moisture from the southern North Pacific due to a southward displacement of the Pacific Storm Track and (ii) a decrease in moisture from the North American continent due to decreased continental recycling during the LGM, as corroborated by $^{17}O_{excess}$ measurements[54]. iCESM does not fully resolve the NAM; we therefore cannot rule out NAM-related processes as potential contributors to DH2 $\delta^{18}O$ on glacial-interglacial timescales. Simulated change in $\delta^{18}O$ between LGM and PI ($\Delta\delta^{18}O$) underestimates the observed $\Delta\delta^{18}O$ in the DH2 record by ~1‰ (considering seawater corrections). This may be due to (i) limitations in iCESM to simulate processes related to NAM strength, (ii) lower LGM-PI temperature differences in iCESM simulations ($\Delta5\,°C$) relative to the true magnitude suggested by proxy reconstructions, and/or (iii) inaccuracies in iCESM ice volume forcing in the southern Sierra Nevada mountain range[55,56], which may alter moisture trajectories and contribute to increased rainout during glacial periods.

## Orbital-scale controls on southern Great Basin climate

To examine the orbital signal of the extended DH2 $\delta^{18}O$ record, we conducted spectral analysis from 582-26 ka. Termination I and the Holocene were not included in our analysis due to poor age control over this interval (see methods). To account for age model errors, 100 individual DH2 $\delta^{18}O$ age models were produced using the OxCal Poisson-Process Deposition model, which uses Bayesian statistics[57] ("Methods"). The absolute-dated DH2 $\delta^{18}O$ timeseries has significant peaks (variance above red noise 95% confidence interval [CI]) in both the precession (23 kyr period) and obliquity (41 kyr period) orbital bands (Fig. 4). A third peak at ~92 kyr is equivalent to the sum of 4 precession cycles and is near the 100 kyr period.

The extended DH2 $\delta^{18}O$ record shows remarkable similarity to other palaeo records that track global-scale climate changes on orbital timescales (Fig. 5). Using the new insights from our iCESM results, potential explanations for this similarity are as follows: (i) fluctuations in greenhouse gas (GHG) concentrations drove regional temperature changes, in which case the DH2 $\delta^{18}O$ would closely follow the atmospheric $CO_2$ record, and (ii) changes in the extent of the Laurentide Ice sheet caused a southward displacement of the Pacific Storm Track which influences the amount of depleted southern North Pacific moisture arriving to DH2, in which case DH2 $\delta^{18}O$ would closely follow marine records that reflect changes in global ice volume. We investigate these and other potential drivers of DH2 $\delta^{18}O$ variability on orbital timescales using three approaches. First, we calculated the phase relationship between DH2 $\delta^{18}O$ and various orbital parameters over the last 500 ka. Second, we determined the phasing of the DH2 $\delta^{18}O$ record relative to multiple palaeo records that are (in)directly linked to mechanisms that may influence DH2 $\delta^{18}O$ on orbital timescales. Lastly, we compared the timing of TII-V in DH2 $\delta^{18}O$ and the aforementioned palaeo records. For each analysis, we examined the phasing of DH2 $\delta^{18}O$ with and without a 880-year groundwater residence time (i.e., adding 880 years to the DH2 $\delta^{18}O$ chronology over the last 500 kyrs), which is the estimated modern residence time from recharge centers to DH and DH2 caves[22–24]. Modern groundwater transit times are considered a conservative maximum, as transit times were likely shorter during glacial periods due to the significantly higher (> 250% at the LGM) recharge to the local aquifer[25].

To assess the long-term drivers of DH2 hydroclimate, we first calculated the phase relationship between DH2 $\delta^{18}O$ and various orbital parameters over the last 500 ka using cross-spectral analysis. The power spectra of 100 individual DH2 $\delta^{18}O$ age models were cross-correlated with the precessional index and obliquity ("Methods", Supplementary Fig. S14). In the precessional period (22–24 kyr), there

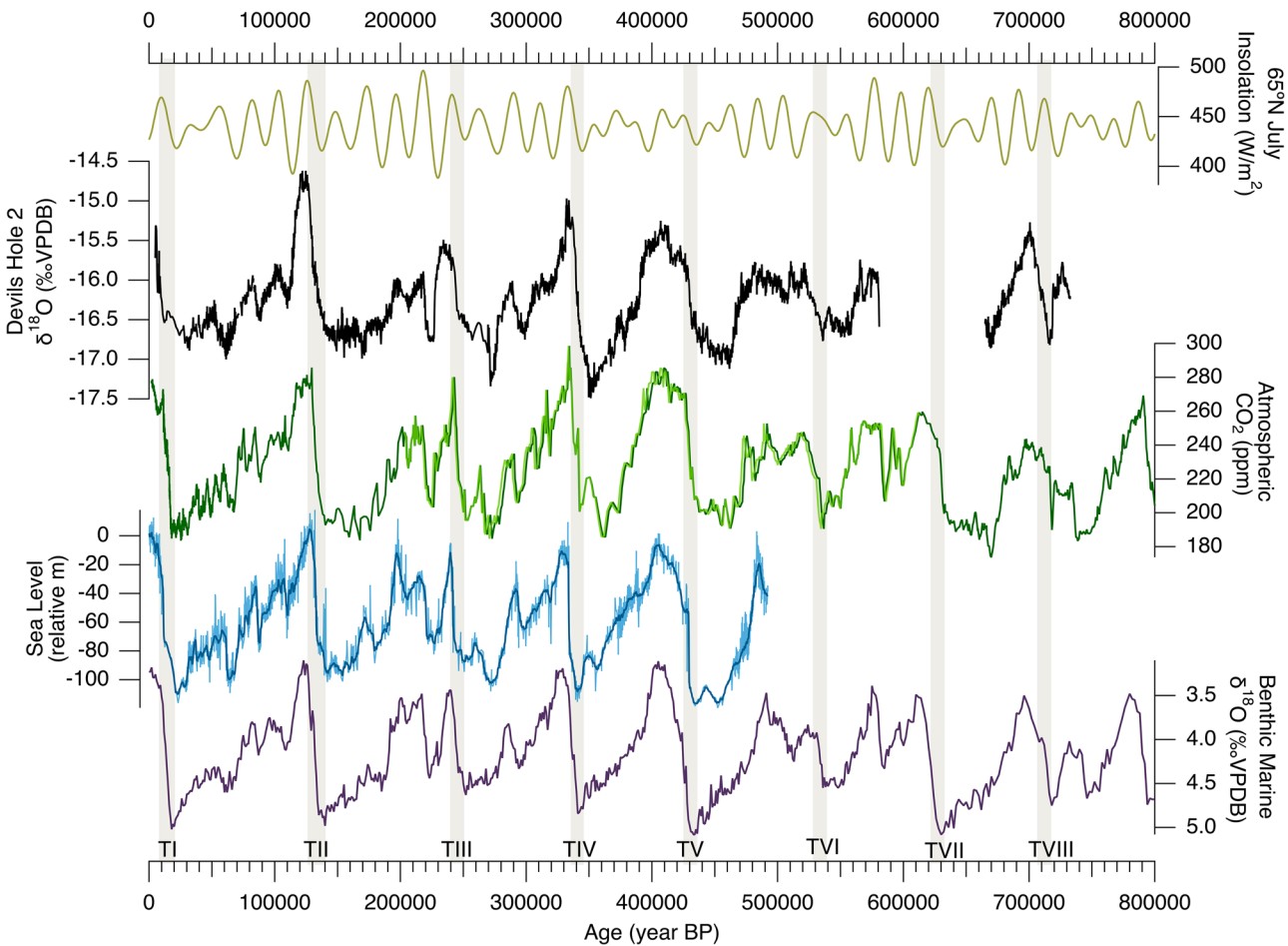

**Fig. 5 | Global context of the Devils Hole 2 $\delta^{18}$O record.** 65°N July insolation (gold)[94], Devils Hole 2 $\delta^{18}$O (black) (*11; this study*), composite atmospheric $CO_2$ record (dark green)[59,65] on the AICC2023 chronology[60] overlain by the composite $CO_2$ (light green)[59] on the WD2014 (0–67 ka;[65]), DF2021 (67–207 ka;[79]), and AICSTAL2024 chronologies (207–600 ka; this study), global relative sea level derived from Red Sea data (blue)[58] with uncertainties (light blue shading), and stacked global benthic $\delta^{18}$O record (purple)[64]. Tan bars indicate the timing of Terminations I–VIII.

is no significant phase lag between the precessional index minimum and insolation maximum at 36 °N or averaged over 50–80 °N, calculated at summer solstice or averaged over the summer months (summer solstice to fall equinox). Summer insolation averaged over 50–80 °N (and 50–80 °S) also has significant power in the obliquity period (38–43 kyr); there is no phase lag between maximum axial tilt and 50–80 °N or °S insolation, calculated at that hemisphere's summer solstice or averaged over the summer months (summer solstice to fall equinox), in the obliquity period. In the precessional period, peaks in DH2 $\delta^{18}$O lagged the precession index minimum, and therefore maximum local and higher latitude NH summer insolation in the precessional period, by a mean of 5.7 kyr (1.4–8.4 kyr, 95% CI, including age model uncertainty), or a mean of 4.8 kyr accounting for maximum groundwater residence time (Fig. 4 and Supplementary Fig. S13). In the obliquity period, peaks in DH2 $\delta^{18}$O also lagged peaks in axial tilt, and therefore the maximum summer insolation of each hemisphere in the higher latitudes in the obliquity period, by a mean of 8.5 kyr (3.8–12.5 kyr, 95% CI, including age model uncertainty) (Supplementary Fig. S13), or a mean of 7.6 kyr accounting for maximum groundwater residence time. DH2 $\delta^{18}$O's 95% confidence interval phase lag range to orbital forcing is an artifact of DH2 age model uncertainties (Supplementary Fig. S14), uncertainty inherent to cross-spectral analysis with relatively short records, and the non-uniformity of climate records in each frequency domain through time. Feedback processes and/or changing groundwater transit times may also contribute to

variable DH2 $\delta^{18}$O lags relative to potential forcings over the last six glacial-interglacial cycles.

Next, we performed the same cross-spectral analysis on an absolute-dated relative sea level (RSL) curve that reflects changes in global ice volume[58] and atmospheric $CO_2$ concentrations[59,60]. The records' 95% CI phase lag range relative to orbital forcing underscores the complex interplay between Earth's orbital variations and ice sheet growth and atmospheric $CO_2$[61]. The multi-thousand-year phase lag of maximum DH2 $\delta^{18}$O, ice volume, and atmospheric $CO_2$ to summer insolation maximum in the precessional phase overlaps between the three records (Fig. 4). This overlap is not unique to DH2, as shown by the Nevada stalagmite $\delta^{18}$O composite (Leviathan record)[13,14] in Fig. 4. We therefore cannot rule out North American ice volume (through the mechanisms outlined above) nor temperature changes (through GHG radiative forcing) as potential drivers of southern Great Basin $\delta^{18}$O variability at orbital timescales.

In addition to cross-spectral analysis, we also investigated phasing of records using a lagged correlation method. Here, climate records were linearly shifted in time relative to the DH2 $\delta^{18}$O timeseries until a maximum correlation coefficient (*r*) was achieved ("Methods"). The records investigated were global atmospheric $CO_2$ concentrations[59], tropical east Pacific SSTs[62], two global RSL records, and the LR04 benthic marine $\delta^{18}$O stack as a proxy for global climate and ice volume[58,63,64]. Resulting maximum correlation coefficients and lags are listed in Supplementary Table S6. The highest degree of correlation

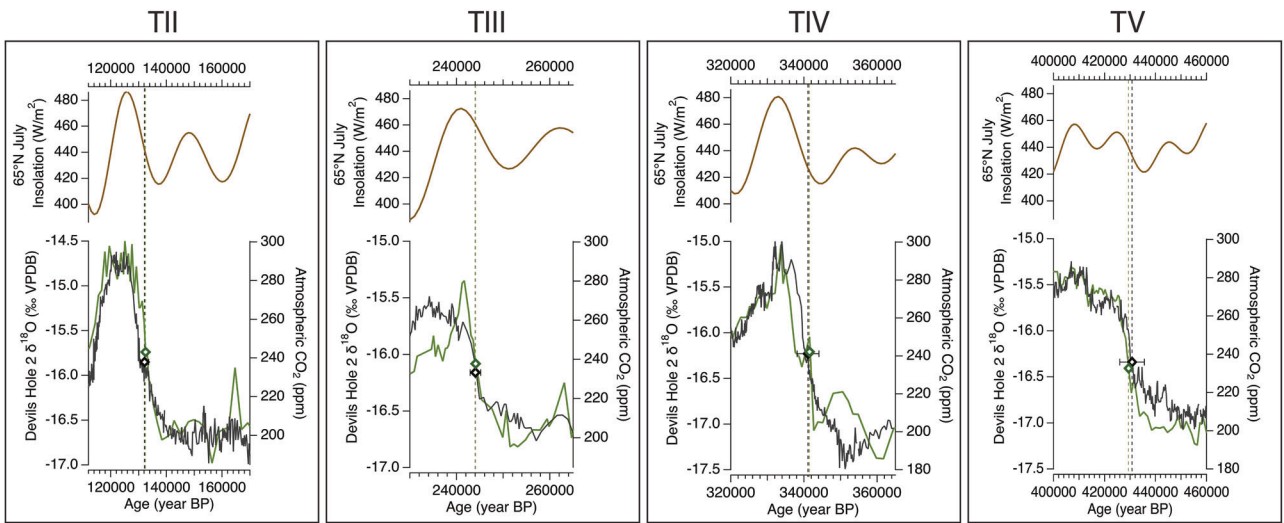

**Fig. 6 | Glacial terminations TII-TV recorded in Devils Hole 2 (DH2) δ¹⁸O.** From top: 65°N July insolation (gold)[94], DH2 δ¹⁸O (black) during T-II[11] and Terminations III−V (this study), compiled atmospheric $CO_2$[66–68] on the AICSTAL2024 chronology (green) (this study). Diamonds and dashed lines represent the respective timing of termination midpoints on the DH2 (black) and $CO_2$ (green) records. Black horizontal bars represent the 2σ age uncertainties of the DH2 age model at each midpoint (black diamond).

($r = 0.87$; $p < 0.01$) occurs between DH2 δ¹⁸O and atmospheric $CO_2$[59,65–68] on the ice core AICC2023 chronology[60] (Supplementary Fig. S12). The same high degree of correlation ($r = 0.87$; $p < 0.01$) is found using an ice core timescale synchronized to absolute-dated Asian speleothem records (AICSTAL2024; see "Methods"). Leads and lags in the $CO_2$ records were < 300 years relative to DH2 δ¹⁸O, results within error of the cross-spectral analysis results. DH2 leads $CO_2$ on the ice core AICC2023 chronology by 650 years when accounting for maximum groundwater residence time (Supplementary Fig. S12). The second highest degree of correlation ($r = 0.77$; $p < 0.01$) occurs between DH2 δ¹⁸O and the absolute-dated RSL curve[58], which lags DH2 δ¹⁸O by 1700 years, which is in contrast to the cross-spectral analysis results. Climate records whose chronologies are on or partially tied to LR04[63,64], including tropical East Pacific SSTs ($r = 0.70$; $p < 0.01$)[62], range in maximum correlation coefficients ($r = 0.70$-0.82; $p < 0.01$) and lag relative to DH2 δ¹⁸O by a range of < 1000 to 5400 years. Respective changes in phasing after accounting for maximum groundwater residence time are shown in Supplementary Fig. S12.

Lastly, we investigated potential drivers of DH2 δ¹⁸O during periods of abrupt warming and drying. To do so, we determined the midpoint of TI-V in DH2 δ¹⁸O and the aforementioned records on their individual chronologies. The midpoint of the ascending limb of DH2 δ¹⁸O during TII (132.15 ± 1.5 ka) was previously calculated by ref. 11. Using the same approach, we calculate the midpoint associated with TIII (244.0 ± 1.1 ka), TIV (341 ± 3 ka), TV (430 ± 6 ka), TVI (529 ± 6 ka) and TVIII (708 ± 8 ka) of the DH2 δ¹⁸O record (Supplementary Table S7; see methods). The maximum groundwater residence time (880 yrs) falls within chronological uncertainties of each midpoint and is therefore not considered in this approach. As shown in ref. 11, multiple growth rate changes in core D during TI and the Holocene resulted in poor age control (up to 18% relative 2σ uncertainty between 18-10 ka). For this study, we focus on TII-TV during which the DH2 record has the best age control (≤ 1% relative 2σ uncertainties) (Fig. 6). The termination midpoints of DH2 δ¹⁸O and selected climate records are listed in Supplementary Table S5. The greatest agreement in the timing of TII-TV midpoints occurs between DH2 δ¹⁸O and atmospheric $CO_2$ on the AICSTAL2024 chronology (Fig. 6). DH2 δ¹⁸O midpoints lagged $CO_2$ by an average of ~100 years, ranging from -0.5 to +0.5 kyr, where negative values indicate that DH2 δ¹⁸O lags $CO_2$. Following atmospheric $CO_2$, the absolute-dated RSL curve shows variable leads and lags of termination

midpoints, with RSL rise lagging DH2 δ¹⁸O by an average of 2.7 kyr (− 0.4 to 7.2 kyr, where negative values indicate that DH2 δ¹⁸O lags RSL) during TII-V.

Overall, our phasing and midpoint analyses suggest that DH2 δ¹⁸O bears the closest structural and temporal similarity to atmospheric $CO_2$ variability, both when calculating an average over the entire record and over short periods of abrupt warming and drying. In the event of a termination, rising $CO_2$ concentrations would warm eastern North Pacific and western North America regions which, in alignment with our iCESM results, would (i) enrich the δ¹⁸O of water vapor from moisture sources, (ii) decrease land-sea temperature gradients (decreasing rainout efficiency along moisture trajectories), and (iii) increase evaporation from the North American continent, collectively resulting in an enrichment of δ¹⁸O moisture arriving to DH2. Phasing using the lagged correlation method and midpoint analysis shows that, on average, DH2 δ¹⁸O leads changes in global ice volume. This suggests that the modulation of the Pacific Storm Track by North American ice sheets may have contributed to (e.g., amplified), but was not a primary driver of, changes in DH2 δ¹⁸O on orbital timescales. Another example of the close coupling between DH2 δ¹⁸O and $CO_2$ is at the onset of terminations. The timing of DH2 δ¹⁸O rise closely aligns with the onset of $CO_2$ rise associated with TII-V, including the "early" rise in DH2 δ¹⁸O and $CO_2$ at the onset of Termination IV (358-345 ka), which is absent from global ice volume records (Fig. 5).

In total, results from two of the three approaches used here suggest that temperature changes are the primary driver of DH2 δ¹⁸O variability on orbital timescales, with secondary drivers stemming from atmospheric circulation changes due to North American ice sheet extent. Both mechanisms outlined here are consistent with orbital forcings (i.e., Milankovitch theory). Changing boundary conditions may amplify or dampen the relative influence of a particular driver on DH2 δ¹⁸O over time, while variable groundwater transit times may vary relative leads and lags.

### Orbital-scale environmental changes in southern Nevada
We now evaluate the extended DH2 record of δ¹³C, a proxy for vegetation type and density in the high-alpine recharge centers of DH2[69]. The DH2 record confirms a majority of the δ¹³C features in the original DH studies, including the magnitude of DH δ¹³C variability[69]. The DH2 δ¹³C timeseries is inversely related to DH2 δ¹⁸O (Fig. 2), such that

enriched DH2 $\delta^{13}C$ broadly corresponds to periods of depleted DH2 $\delta^{18}O$ and high palaeo water tables[27,28]. Cross-spectral analysis shows that DH2 $\delta^{13}C$ leads DH2 $\delta^{18}O$ by an average of ~2 kyrs. Dissolved inorganic carbon (DIC) residence times cannot be reliably calculated in the DH2 aquifer due to extensive carbon exchange between groundwater and carbonate bedrock[23,69] while dissolved organic carbon (DOC) $^{14}C$ measurements suggest that DOC transit times are similar to but longer than groundwater transit[23]. It is therefore surprising that the $\delta^{13}C$ signal arrives before $\delta^{18}O$, as first pointed out by Coplen et al.[69], and suggests alternative forcings from $\delta^{18}O$ of precipitation. The DH2 $\delta^{13}C$ record's spectral structure shows significant peaks in the precession band (23 kyr period) and the 100 kyr period, but the small peak in the obliquity band is indistinguishable from red noise (Supplementary Fig. S8). DH2 $\delta^{13}C$ variations in the precessional band are in-phase with seasonal insolation during the warm growing season (between summer solstice and fall equinox) in the Great Basin (Supplementary Fig. S13).

Coplen et al.[69] first proposed that variations in DH2 $\delta^{13}C$ result from the isotopic composition of DIC in groundwater that is primarily generated in soils at recharge zones. They argued that DH2 $\delta^{13}C$ largely reflects the $\delta^{13}C$ of soil air, which varies inversely with vegetation density and primary productivity[69]. Vegetation type may have also contributed to the change in DH2 $\delta^{13}C$: the upper elevations (3650-3400 m) of the Spring and Sheep Mountains are dominated by arctic-alpine shrub and grassland[36], which contain higher $\delta^{13}C$ values relative to the forest cover at lower elevations. Packrat middens suggest that ecotones in southern Nevada shifted as much as 1000 m lower in elevation during the last glacial maximum[70], extending the arctic alpine grasslands cover to the upper ~2000 m of Spring and Sheep Mountains during glacial maxima. Our phasing analysis reveals that DH2 $\delta^{13}C$ is in-phase with strong seasonality, which, following the interpretation of ref. 69, suggests that increased vegetation density and primary productivity is positively correlated to warm Northern Hemisphere summers.

As observed in the original DH record, DH2 $\delta^{13}C$ reached its lowest values in the first half of interglacial periods (Fig. 2). However, the timing of the prominent troughs are shifted by our new chronology to younger values that coincide with peak boreal summer insolation: 426 ± 9 ka (MIS 11e), 335 ± 4 ka (MIS 9e), 240.1 ± 1.5 ka (MIS 7e), and 125.8 ± 0.4 ka (MIS 5e). The timing of prominent lows coincides with periods of regional warmth, as suggested by DH2 $\delta^{18}O$, which supports the hypothesis that DH2 $\delta^{13}C$ lows represent periods of dense vegetation cover and high primary productivity in the high-elevation recharge centers to the local aquifer[69]. Prominent lows are followed by an abrupt reversal in $\delta^{13}C$, despite relatively high DH2 $\delta^{18}O$ values throughout the end of each interglacial period. For example, DH2 $\delta^{18}O$ reaches maximum interglacial values during MIS 5e at approximately 127 ka before plateauing for ~7 kyrs[3,13,14], whereas DH2 $\delta^{13}C$ reverses towards higher values at 126 ka. Reversals in DH2 $\delta^{13}C$ coincide with a rapid lowering of the DH2 water table from glacial high stands (+9-10 m) to levels similar to today[27,28]. The DH2 $\delta^{13}C$ reversal associated with MIS 5e, 7e and 9 occurs when the palaeo water table reaches below a threshold of +3.7 m (Fig. 2), which is equivalent to +52% recharge relative today[25]. Declining effective moisture, coupled with warm temperatures in the first half of interglacial periods, triggered a loss of vegetation density in the high-elevation recharge centers of DH2. This loss continued throughout the latter half of each interglacial. In total, the extended DH2 $\delta^{13}C$ record suggests that seasonality is the dominant driver of orbital-scale environmental change in the highlands of southwest Nevada, with a tipping point in effective moisture (<50% above modern) that results in a rapid and unilateral decline in primary productivity during warm interglacials.

In summary, the phasing of DH2 $\delta^{18}O$ timeseries suggests that temperature-related processes are dominant drivers of orbital-scale $\delta^{18}O$ variability in southern Great Basin precipitation, consistent with

iCESM model outputs. Global ice volume lags DH2 $\delta^{18}O$ on average, suggesting that mechanisms linked to the North American ice sheets (e.g., shifting storm track) contributed to, but were not the primary driver of, DH2 $\delta^{18}O$ variability at these timescales. In contrast, the DH2 $\delta^{13}C$ timeseries is in-phase with seasonal insolation during the warm growing season (between summer solstice and fall equinox). Prominent $\delta^{13}C$ lows, indicating high primary productivity, coincide with peak boreal summer insolation during the last six interglacial periods. A rapid decrease in vegetation density coincides with warm interglacial temperatures and <50% greater recharge relative to today, suggesting a tipping point for local environmental decline. This study sheds new light on the relationship between temperature, moisture balance, and vegetation in the southern Great Basin on orbital timescales. Our results underscore the link between increased $CO_2$ concentrations and regional warming[18,19,71,72], which is expected to contribute to reduced effective moisture in the Great Basin over the coming century[16,20].

## Methods

A 670 mm-long core was drilled in the hanging wall of Devils Hole 2 (DH2) cave at +1.8 m r.m.w.t. (core D). The core was cut longitudinally and inspected for growth hiatuses and petrographic changes. The first 654 mm of the core is calcite; the last 16 mm of the core is bedrock. The core consists of two types of calcites: folia and mammillary calcite. The latter precipitates subaqueously, while folia forms shelf-like formation near the air-water boundary[27,34]. The presence of folia in the core is an indicator of a palaeo-water table near +1.8 m r.m.w.t. at the time of deposition. Folia is present in core D at 77.7–97.4 mm, 171.4–199.2 mm, 209.4–229.0 mm, and 305.0–323.0 mm (distances are reported from the top of the core, i.e., the modern cave wall surface). U-series dating at folia boundaries confirms four growth hiatuses in the record, during which the water table was below +1.8 m r.m.w.t. and deposition ceased. These time periods coincide with interglacials Marine Isotopes Stages (MIS) 5, 7a, 7e and 9. In order to construct a continuous record, these growth hiatuses were filled with data from a core collected in DH2 from a lower elevation (core P, -1.6 m r.m.w.t.), which grew continuously during these times. A low-resolution (2 mm) stable isotope record from core P was sampled to identify the interglacial DH2 $\delta^{18}O$ peaks (Supplementary Fig. S1). Higher resolution (0.2 mm) stable isotope values were measured at each interglacial period identified on core P (Supplementary Fig. S1 and Supplementary Table S4). The $\delta^{18}O$ values of core P are offset by −0.1 (MIS 5e), 0 (MIS 7a), −0.1 (MIS 7e) and −0.25‰ (MIS 9) to be visually aligned with the existing data (Supplementary Table S4). A constant growth rate for each of the four core P deposition phases is assumed, thus, the added core P stable isotope data are evenly spaced on the core D depth scale. Finally, a growth hiatus in core D was discovered between 587.4 and 589.0 mm dated to 665-580 ka[29], and has no known corresponding growth phases in lower elevation cores.

### U-series dating

Mammillary calcite was $^{230}Th$-$^{234}U$ and $^{234}U$-$^{238}U$ dated at regular intervals along the core, totaling 123 U-series ages. Folia calcite behaves as an open system and therefore cannot be directly dated[11]. Results for 100 $^{230}Th$-$^{234}U$ ages were published by refs. 11, 35, and 29 (Supplementary Table S1). In addition, 10 $^{234}U$-$^{238}U$ ages were measured by ref. 29 (Supplementary Table S2). For this study, one $^{230}Th$-$^{234}U$ age and three $^{234}U$-$^{238}U$ ages were additionally measured at the University of Minnesota using methods identical to the aforementioned publications (Supplementary Tables S1, S2). Calcite powders (30–50 mg) were hand drilled at regular intervals using 0.3-0.4 mm carbide-tipped drill bits and spiked with a mixed $^{233}U$-$^{236}U$-$^{229}Th$ spike. Spiked samples were dissolved, centrifuged, and loaded into anion exchange columns following the methods described in ref. 73. U and Th aliquots were extracted and measured using a ThermoFisher Neptune Plus MC-ICP-MS via a secondary electron multiplier on peak-jumping mode[73,74].

Chemical blanks were measured with each set of 10–15 samples and were found to be negligible (< 50 ag for $^{230}$Th, <100 ag for $^{234}$U, and < 1 pg for $^{232}$Th and $^{238}$U).

## Stable isotopes

Samples for stable isotope measurements were micromilled continuously at 0.1–0.2 mm intervals along the core axis. Samples between 0–158.4 mm are presented in ref. 11. Between 158.6–654.0 mm, 2505 new stable isotope samples were micromilled for this study at 0.2 mm intervals. Calcite powders were measured at the University of Innsbruck using a Delta V plus isotope ratio mass spectrometer interfaced with a Gasbench II. Values are reported relative to VPDB with 1-sigma precision of 0.08‰.

## Age model

An age model was calculated by ref. 11 the first portion of the core (0–158.4 mm) spanning 4.90–204.2 ka. For this study, a second age model was calculated between 153.0–654.0 mm. Nine of the 49 $^{230}$Th-$^{234}$U ages between 153.0 and 654.0 mm were out of stratigraphic order within uncertainties and therefore excluded (Supplementary Table S3). The 153.0–654.0 mm age model was calculated using the Bayesian statistical software OxCal version 4.2 under deposition sequence "P" with k-parameter set to 0.1[75]. The positions of growth hiatuses, including folia calcite, were incorporated into the age model as growth boundaries.

## iCESM Model simulations

We evaluate previously published simulations using the Community Earth System Model version 1.3 with water isotopologue tracking of oxygen and hydrogen in the atmosphere, land, ocean, sea ice, and runoff components[52]. These simulations were performed as the isotope-enabled Transient Climate Experiment for the last deglaciation (iTRACE), as published in refs. 53,76, and we refer to those studies for details on the simulations. Moisture tagging studies were done at the LGM and PI, in which evaporating water and its isotopologues are tagged with its origin, allowing it to be traced through the hydrological cycle[53].

## AICSTAL2024 ice core gas chronology

As part of this study, we have created a new 207–600 ka ice core gas chronology ('AICSTAL2024') by tuning the AICC2012 gas chronology to Chinese stalagmite U-Th dated $\delta^{18}$O tie points, based on the relationship between shifts in Chinese stalagmite $\delta^{18}$O and atmospheric CH$_4$[77]. We combine the 207–600 ka AICSTAL2024 chronology with WD2014 (0–60 ka[78]) and DF2021 (67–207 ka[79]) for a full 0–600 ka ice core gas chronology.

The AICSTAL2024 chronology was constructed as follows: Tie points (Supplementary Fig. S9 and Supplementary Table S5) were selected between the Dome C (EDC) CH$_4$ record[80] on the AICC2012 age model[81] and the East Asian Summer Monsoon stalagmite $\delta^{18}$O record ("EASM")[82]. Tie points were also selected between the Vostok CH$_4$ record[67] on the AICC2012 age model and the Asian Monsoon stalagmite $\delta^{18}$O record[82] (Supplementary Fig. S9 and Supplementary Table S5). The tie points were then uploaded into the QAnalySeries program[83,84] to create an EASM-tuned gas age-depth model for EDC and Vostok. Piecewise cubic spline interpolation (Matlab function) was used to assign ages to the EDC and Vostok CO$_2$ samples using the EASM-tuned gas age-depth model. The AM-tuned EDC and Vostok CO$_2$ records were combined to create a 207–600 ka composite CO$_2$ record, following the same structure as[59]. Differences between CO$_2$ records on the AICC2021 and AICSTAL2024 chronologies are shown in Supplementary Fig. S10.

## Spectral analysis

Spectral analysis was completed on the DH2-D $\delta^{18}$O record from 582-26 ka. Note, Termination I and the Holocene are not included in the spectral analysis, due to the multiple growth rate changes and possible poorly constrained growth hiatuses in the DH2-D record during these periods. Further, the section of the record older than 582 ka is not included in the spectral analysis because of the large age gap between 587.4 and 589.0 mm depth (> 80 kyr duration).

Spectral analysis of DH2 $\delta^{18}$O was performed using (i) REDFIT[85,86], (ii) the Lomb-Scargle method[87,88], and (iii) the Multi-taper method (MTM)[89] (Supplementary Figs. S8, 7). REDFIT[85] and Acycle[90] software were used for spectral analysis, red noise and 95% significance threshold computation. As high-frequency variations in the records are not relevant to this study, for spectral analysis, the records were first transformed to 1000-yr evenly spaced interval records using the bin method. In the REDFIT method, the number of segments with 50% overlap (n$_{50}$) was set to 2, a Hanning window type was chosen to avoid spectral leakage, and the red noise rho (ρ) value was set as the average of the one-lag autocorrelation and the square root of the two-lag autocorrelation of the record. For the MTM method, the default three 2π prolate tapers were used with no zero padding. The robust AR[1] noise model[91] and the Power Law model[92] were used to model noise and 95% significance, and are compared in Supplementary Fig. S7.

To account for age model errors, 100 individual age models produced as part of the OxCal v4.4 variable deposition age model program were extracted and analyzed individually[57,75,93]. To reduce computing time to a manageable level, OxCal was first run without interpolation to produce an age model with sequential ages with 95% confidence ranges. The model was then run a second time, inputting the modeled ages from the first run so that we were able to extract 100 age models with interpolation of ~ 5 measurements per mm. The initial k-parameter was set to 0.01 mm$^{-1}$, with an allowable k parameter range between k0*0.01 to k0*1000. OxCal input scripts for step 1 and step 2 are provided in the Supplementary Materials.

Spectral analysis was repeated 100 times on evenly spaced 1000-yr binned $\delta^{18}$O associated with each of the 100 age models using the REDFIT software with a prescribed red noise ρ value for each record. To plot all 100 records' power spectrum in a composite plot, each power spectrum was normalized to its corresponding red noise 95% false alarm record (Supplementary Fig. S8).

Cross-spectral analysis between the evenly spaced 1000-yr binned DH2 $\delta^{18}$O record and orbital precession (esinω where ω is the longitude of perihelion measured from the moving vernal point and e is the eccentricity of Earth's orbit around the sun) and obliquity[94,95] was computed using REDFIT-X[96]. The analysis was performed on the DH2 mean age model (Fig. 4b) and repeated 100 times using the 100 age models (Supplementary Fig. S14). For cross-spectral analysis with orbital precession, phase lag 95% confidence intervals at periods with significant coherence (> 0.8) and within the precessional period (22–24 kyr) were averaged for a final precessional-period phase lag 95% CI. For cross-spectral analysis with orbital tilt, phase lag 95% confidence intervals at periods with significant coherence (> 0.8) and within the obliquity period (38–43 kyr) were averaged for a final obliquity-period phase lag 95% CI. To account for maximum groundwater residence time at DH2, 880 years were uniformly added to the DH2 chronology, reducing the calculate phase lag (Fig. 4b).

## Lead-lag analysis

Correlation coefficients between DH2 $\delta^{18}$O and the first 500 kyr of select climate records were obtained using a lagged correlation method in MATLAB. In this method, each record was linearly shifted in time relative to the DH2 $\delta^{18}$O timeseries until a maximum correlation (r) was achieved. Supplementary Fig. S12 shows probability density curves fitted by the Gaussian distribution. MATLAB scripts are provided in the Supplementary Information (Supplementary Code 1). To account for maximum groundwater residence time at DH2, 880 years were uniformly added to the DH2 chronology.

## Midpoint analysis

Following previous studies, we define Devils Hole "midpoints" as the first point at which the $\delta^{18}O$ record exceeds $(\delta^{18}O_{max} + \delta^{18}O_{min})/2$, where $\delta^{18}O_{max}$ is the maximum (enriched) $\delta^{18}O$ value and $\delta^{18}O_{min}$ is the minimum (depleted) $\delta^{18}O$ value of the rising branch of $\delta^{18}O$ during terminations[11].

## Data availability

The Devils Hole 2 $\delta^{18}O$ and $\delta^{13}C$ data is available in the Supplementary Information. Source data are provided in this paper.

## Code availability

MATLAB code for lagged correlation analysis is available in the Supplementary Information (Supplementary Code 1).

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

## Acknowledgements

This research was funded in part by the Austrian Science Fund (FWF) 10.55776/P32751 and 10.55776/P26305 to C.S., the National Science Foundation project numbers 1602940 and 2202913 to R.L.E., National Natural Science Foundation of China project number 41888101 to H.C., National Science Foundation project number 2102944 to C.B., and the Heising-Simons Foundation grant number 2022-3756 to K.W. Special thanks to Ikumi Oyabu and Nick Scroxton for data sharing and scientific input. This research was conducted under research permit numbers DEVA-2010-SCI-0004 and DEVA-2015-SCI-0006 issued by Death Valley National Park. We thank K. Wilson for assistance in the field and M. Wimmer and X. Li for assistance in the laboratories. For open access purposes, the authors have applied a CC BY public copyright license to any author accepted manuscript version arising from this submission.

## Author contributions

K.W., G.M., and S.S. conducted sample measurements and analyses. S.C. performed spectral analysis. C.H. and C.B. contributed and analyzed climate modeling results. M.W. provided modern climate data analysis. Y.D. conceptualized the project and secured sampling permits. R.L.E., H.C., and C.S. provided scientific guidance and funding.

## Competing interests

The authors declare no competing interests.
