## [Transparent Peer Review file · Nature Communications]

Controls on the southwest USA hydroclimate over the last six glacial-interglacial cycles

Corresponding Author: Dr Kathleen Wendt

Version 0:

Reviewer comments:

Reviewer #1

(Remarks to the Author)

Review of Wendt et al. "Controls on the southwest USA hydroclimate over the last seven glacial-interglacial cycles" for Nature Communications, July 2024

Summary

Wendt and colleagues present an update to the U-Th dated Devils Hole d18O (and d13C) record that includes filling in several important missing gaps and extending the record back significantly beyond 700 kyrs. This site and record is hugely important in understand western US paleohydrology in the Quaternary and this work (and the previous work from this group) is a tour de force. I am not an expert on the spectral analyses conducted, as such I commented primarily on the geochemical, geological and paleoclimate aspects of the work. There are limitations that need to be addressed, but overall I am supportive with moderate revisions of eventual publication of this exciting and important work.

Major Comments

The following are major comments that require some addressing but overall nothing here detracts from the overall story and conclusions of the work and I'm sure will be easily handled by the authors:

1) Figure 4 and analysis related to this:

- If Leviathan is going to be used on Figure 4 it should be shown on Figure 3 in my opinion.
- 239-242 and 255-257: Some sort of definition what Milankovitch and this work considers "near" ice sheets. The southern Great Basin is actually pretty far from the edge of the Laurentide/Cordilleran ice sheet.
- What dictates the size of the slice in Figure 4? Is it some sort of implied error bar?

2) Generally speaking I found the references to not be sufficient when I thought carefully about the points the authors were making with respect to the Great Basin lakes. I have made numerous suggestions in this regard below both regarding lake records that do record multiple glacial-interglacial cycles and chronology and compilation efforts over recent years. I hope the authors will consider using more appropriate references for the pluvial lake observations and sediment core based records.

3) Confounding precipitation d18O as recorded here in groundwater d18O with regional moisture balance changes over glacial-interglacial timescales. As noted in specific comments below, I did not find the writing clear and all places where broader regional hydroclimate is being interpolated should be scrutinized with respect to phrasing.

4) Groundwater residence times feeding the DH/DH2 record. The authors acknowledge the residence time being ~800+ years early on in the paper, but I was surprised that:

- This was not accounted for in the lead-lags that were analyzed, especially given that these leads and lags would be variable given the recent groundwater modeling by Jackson et al. (2023) (a paper which several of the authors here are also on).
- Wendt et al. reference 74 and the observed variations in (234U/238U) are significant in that the variations perhaps even allow us to understand the groundwater paleohydrology from the DH cores. The authors did not mention or comment on this whatsoever despite this clearly being relevant.

Minor Line-by-Line Comments

48: "Full-record analysis" – this is too much jargon. I realize the need for brevity in this journal's format but please clarify this sensen.

65: The references 1-4 included here other than Lowenstein et al. 1999 are not suitable. They only cover the last glacial and

deglaciation. I would suggest instead citing work from Owens Valley (from the >800 kyr OL-92 core), Searles (recent work by Peapple/Stroup and older work by Lowenstein), Manix (Reheis et al., 2012) and Bonneville/Bear Lake (cores that go back 2-3 glacial cycles) that shows “repeated expansion” of the lakes. I understand that the references used here are somewhat canonical in the pluvial lake literature but to make the point being made in this sentence of repeated pluvial lakes over glacial-interglacial cycles some other references are needed. You might also consider citing Reheis et al. (2014) and the recent paper by Lowenstein et al. (2024) on Death Valley.

69: Remove “the” before “Milankovitch”

84-86: these references, 12-15, are good references but largely only acknowledge circulation change. But the sentence is focused on the expansion of pluvial lakes and the attribution of temperature (via evaporation) vs moisture delivery. I suggest you cite work by McGee et al. (2018) and Ibarra et al. (2018) here as the role of evaporation on the expansion of pluvial lakes is also non-trivial.

83-93: The rest of this paragraph is largely about precipitation d18O (or dD) rather than hydroclimate (P-E and/or temp vs. precipitation) overall I think the authors could do a much better job defining what they mean by “hydroclimate change” – do they merely mean the d18O (or dD) of precipitation? Rather than effective moisture and/or regional water balance and/or moisture divergence and convergence (as implied by some of the papers that are cited in this paragraph)? More precise writing is needed here.

99: Confusing that then in the next paragraph it is plural (since DH and DH2 are separate caves but the same groundwater system) – it would be good to be consistent/clarify.

140-145: I don’t disagree with this overall conclusion at the broad scale, but I will mention that the authors are brushing over a lot of work done using modern, higher resolution back-trajectory analysis from nearby that also shows the impact of the high southern Sierras, that presumably would be amplified by the Sierra Nevada ice cap leading to wrap around effects and increased moisture rainout during glacials. See the back trajectory analyses by Lechler and Galewsky (2013, *Geology*), Mix et al. (2019, *EPSL*) and Gagnon et al. (2023, *GSAB*). My suggestion is to expand on the sentence from line 145-147 and acknowledge this issue.

165-166: How much excess 230Th is there in core D in the Holocene or at that depth today? Would be best to be specific rather than just “not significantly affected”

194-195: Note that Bhattacharya et al. (2024, *Elsevier*) commented on this and the apparent magnitude of dD vs d18O variations (accounting for meteoric water line slopes).

282-284: Add refs for these records that were used.

305-307: Have the authors considered generating a LGM to Holocene record from one of the other cores near D to get around this problem? (I realize this is outside the scope of this paper/study).

325: define fast? Also what about the 900 year water residence time of the system?

336-338: Fix reference formatting, the full list of authors is not needed here.

352-353: With all due respect to the author of reference 65, the geochronology and compilation literature from the last decade or so of paleo lake high stand datasets should be cited as that is the actual geological data. References such as - Munroe and Laabs (2013); Ibarra et al. (2014); Reheis et al. (2014); McGee et al. (2018); Hudson et al. (2019) - rather than a modeling paper. I do think somewhere reference 65 should be included, perhaps when other similar work by Oster or others is cited.

353-356: Similar to my comments in the introduction the authors seem to be confounding d18O variations with P-E/effective moisture in these sentences.

358-360: Isn’t this entirely expected given the 800+ year residence time of water in the system?

380-400: What is the residence time of DIC in the groundwater system relative to the water itself? Is this known from radiocarbon measurements for example or not possible due to carbonate bedrock contributions to DIC?

410-411: This phrasing is a bit over the top.

411: The paper just ends – this is a bizarre ending to a very worthwhile study. While the format/word limit must be coming into play here the reader is left with no summary or higher-order connection being made and (bizarrely) is left on a discussion of the d13C record that was not mentioned until the last few paragraphs.

446: I was very surprised not to see some discussion of the Wendt et al. (23U/238U) variation observations relative to the groundwater table and newly compiled d18O record.

546-550: d18O not formatted correctly.

Please note that some of the supplementary figures are not uniformly formatted - I would suggest this be fixed in revision.

Reviewer #2

(Remarks to the Author)

The manuscript by Wendt et al. analyzed data from Devils Hole in the Great Basin to investigate a ~700ky hydroclimate record. The study shows that the Great Basin’s drylands experienced significant hydroclimate shifts throughout the Quaternary and that these shifts are linked to large-scale climate changes in the eastern North Pacific and western North America. The study also shows that Devils Hole $\delta^{18}\text{O}$ correlates with variations in atmospheric CO₂ levels and Northern Hemisphere ice volume across seven glacial-interglacial cycles. Particularly, increases in $\delta^{18}\text{O}$ during glacial terminations were associated with rising CO₂, potentially indicating a rapid climate response to temperature changes and radiative forcing. The manuscript is well-written, and the results are very interesting and well-detailed, which allowed me (a non-expert on Great Basin hydroclimate and spectral analysis) to understand the work and its implications. I have few major and minor comments that I hope are found useful by the authors.

Major comments:

Models shown in Figure 1B suggest that northern North Pacific sources of moisture are going to be depleted compared to southern North Pacific sources. If during the LGM southern North Pacific sources of moisture became more prominent due to a southern displacement of the jet, wouldn’t that result in more positive $\delta^{18}\text{O}$?

Lines 258-261: The authors argue that there is not a “simple 1:1 link between North American ice volume and DH2 $\delta^{18}O$ on orbital timescales, but instead for changing boundary conditions and/or multiple feedback processes that act in tandem to vary the lag times of DH2 $\delta^{18}O$ relative to maximum summer insolation.” My understanding is that ice volumes also lag summer insolation by approximately 5,000 years and that ice sheet response is a result of the complex interplay between the Earth's orbital variations, the thermal inertia of ice sheets, and the dynamics of the global climate system. This is in some ways shown in Figure 4 and highlights that both ice sheet volume (based on SL curves) and DH2 $\delta^{18}O$ are similar, or at least within age model uncertainty. It is my opinion that, even though a simple link cannot be established, there is no way to statistically separate them. In the end, it is a very complicated system, involving ice dynamics, atmospheric circulation changes, and water recharge lag times that probably complicate the correlation of the $\delta^{18}O$ record with other time series.

A main concern is the relationship established between Great Basin hydroclimate and atmospheric CO₂ concentrations. The authors argue for this relationship by analyzing the records in two ways and show that $\delta^{18}O$ and CO₂ are strongly correlated, but correlation does not always imply causation. The authors state “While continental ice sheets remain a major control of landscape moisture availability in western North America (12-15), which has been shown to partially influence Great Basin $\delta^{18}O$ (15), the phasing of DH2 $\delta^{18}O$ also suggests a fast-acting climate response to regional temperature changes driven by greenhouse gas concentrations.” It is my opinion that there is a strong correlation between all the proxy records analyzed. Authors should not rule other correlations as potential hydroclimate forcings out just because CO₂ and $\delta^{18}O$ have a slightly stronger correlation.

The entire section of the manuscript related to “Terminations in the Great Basin” is confusing and vague. The authors argue that there is a link between the maximum rate of change in $\delta^{18}O$ and speleothem records from China. However, the synoptic scale changes suggested to explain the correlation between the East Asian Monsoon and Great Basin hydroclimate are vague. Why is the maximum rate of change used? What does this rate of change represent? The authors state that the “link between the East Asian Monsoon and Great Basin hydroclimate may be as follows: meltwater entering the North Atlantic alters ocean and atmospheric circulation and associated heat fluxes, prompting an enrichment of the East Asian monsoon water isotope (62, 64). Around this time, paleo lake high stands are recorded across the southern Great Basin (65).” but then fail to explain why meltwater entering the North Atlantic Ocean could result in a wetter climate in the western North America. The authors then state “The mechanisms that drive millennial-scale wet events in the Great Basin are debated (66-68), with recent studies suggesting increased delivery of Pacific moisture due to a weakening of the North Pacific Hadley circulation in response to northern high-latitude cooling (69, 70).” but this is now talking about atmospheric circulation in the Pacific Ocean, unrelated to North Atlantic meltwater.

Minor comments:

Line 81: This part is a bit confusing to me as the new DH record is called DH2 but DH2 is also Devils Hole 2.

Line 89-90: Can you please specify what are the challenges in the interpretation of $\delta^{18}O$?

Line 92-93: I find this comment vague. Based on your results, can you add what are the implications of higher atmospheric CO₂ on Great Basin hydroclimate? Would we expect an even drier/warmer climate?

Line 320: Change “...DH2 $\delta^{18}O$ and atmospheric CO₂ consistent with the moisture...” to “...DH2 $\delta^{18}O$ and atmospheric CO₂ is consistent with the moisture...”

Line 322: Change “...storm systems move western North America.” to “...storm systems move towards western North America.” or “...storm systems move into western North America.” Not sure what works here but there is something missing in the original sentence.

Line 325: Change “...phasing of DH2 $\delta^{18}O$ also suggests a...” to “...phasing of DH2 $\delta^{18}O$ with CO₂ also suggests a...”

Line 366: Change “...with an acceleration Northern Hemisphere...” to “...with an acceleration of Northern Hemisphere...”

Figure 1: The width of the trajectories has to be better explained. I understand that it represents clustered trajectories. Is it a percentage of trajectory tracks? Is it just a relative number between each other? There is also reference in the manuscript to particular storm tracks, such as the Pacific storm track. Do all these tracks have names? If so, they should all be added to the figure.

Figure 1. LGM and PI should be defined within the figure caption.

Figure 3: Y axis of Devils Hole $\delta^{18}O$ record should be “Devils Hole 2”.

Figure 5: The dashed line for the CO₂ curve mid-point is not shown.

Figure S3: There is something wrong with the embedded text in this figure.

Figure S9: No tie points are shown. Y-axis delta symbols need to be changed.

Figure S13: There is something wrong with part b. Why do you have months instead of phase lag, such as that shown in part a? The manuscript text discusses lags in obliquity as well, but this is not shown in Figure S13. In lines 387-388, Figure S13b is explained but I think there is something off here. Would it be called differently than $\delta^{13}C$ vs precession index phasing?

Reviewer #3

(Remarks to the Author)

Controls on the southwest USA hydroclimate over the last seven glacial-interglacial cycles

Recommendation: Publish after moderate revision

This manuscript presents one of the longest absolute-dated, terrestrial paleoclimatic records from Devil's Hole, Nevada which extends the original DH record back to 736 ka (the new record is called DH2). The primary proxy is a high-resolution set of oxygen isotope measurements from the calcite and while most of the U-series dates constraining this record (114 in

total) were previously published, this paper provides one new ^{230}Th - ^{234}U date and three new ^{234}U - ^{238}U dates for the older part of the record. The two calcite vein cores comprising the record are taken near the water table in Devils Hole 2, eliminating the offset in U-series ages relative to water depth that provided anomalously young ages in the original DH record. While the oldest part of the record dates to 736 ka, there is a depositional gap from ~580 ka to 660 ka which precludes analysis / interpretation of climatic changes over this interval. The older part of the record is time constrained by the new ^{234}U - ^{238}U dating technique that relies on understanding the past $d^{234}\text{U}$ initial in the DH calcite (discussed in Li et al., 2021).

The results presented here build on those of Moseley et al., 2016, showing that hydroclimatic changes in the southern Great Basin are driven by orbital-scale processes back to 736 ka (with the age gap caveat noted above), and they build on the original DH record with a new, higher-resolution set of $d^{18}\text{O}$ measurements. Using an isotope-enabled GCM, the DH2 record is interpreted in terms of large-scale changes across North America (ice sheets, temperature and evaporation) and Pacific SSTs. Specifically, the southern Great Basin is interpreted to have received higher winter precipitation amounts with that moisture sourced from the southern North Pacific and carried along a southerly displaced storm track.

The long-term record presented here will be of broad interest to the paleoclimate community and the modeling results interpreted in tandem with the oxygen isotope data help further constrain the climate dynamics associated with the dramatic changes in hydroclimate over multiple glacial-interglacial cycles. The detailed statistical analysis that a well-dated record like this allows shows the importance of rising CO_2 levels at glacial terminations as being a key driver of local climate change.

The paper is well-written and the conclusions are important enough to merit publication in Nature Communications. However, before the paper can be published, there are a few minor to moderate revisions, that include citation of another recently published DH2 record by the same group of authors, more detail on the interpretation of the climate model results and further and more detailed discussion of the implications of this work.

First – a significant paper detailing water table fluctuations in Devils Hole 2 was published online July 14, 2024 in Communications earth & environment (“Moisture availability and groundwater recharge paced by orbital forcing over the past 750,000 years in the southwestern USA) by the same group of authors as this manuscript. This paper is not cited nor referred to at all in the current manuscript despite being from the same location and covering roughly the same time interval (750 kyr vs. 736 kyr). I assume the reason for this is that Nature Communications does not allow citations of in press work until it is published. I’ll also note that the proxy records are different – the published paper records water table variations in Devils Hole 2 from a different set of calcite cores (cores M, R, L, K, H, I, all of which are taken from above the current water table in DH2) than the current manuscript (core D which is the primary record with spliced in intervals from core P). Each set of cores has its own unique set of radiometric dates, and no stable isotope work was done in the published paper, thus these are different proxies.

However, this manuscript would benefit from including the results of this work as it covers the same time interval (and in fact includes the 80 kyr gap from the current work), and it supports the orbital scale delivery of more precipitation during glacial periods. Figure S15 shows results from this paper plotted against earlier, shorter records of DH water table fluctuations – back to about 350 ka – so including this most recent work would make this correlation much stronger. I assume now that this other work is fully published, that it can be cited and used to support the current manuscript.

Oxygen isotope results and interpretation. I really like using the isotope-enabled GCM to support this work – it provides a compelling explanation of the circulation changes from glacial to interglacial climates and strengthens the paper overall – especially the finding of source regions in the Pacific changing. However, as I note below, digging into the model results in more detail would help explain why the glacial isotopic values are more negative – and it would help explain why the amplitude of isotopic change in the model is significantly less than in the observations (even as it is in the right direction).

The statistical work on this well-dated record is outstanding. The phase lags to various forcing parameters including precession and obliquity, sea level, and atmospheric carbon dioxide provide many insights into the nature of climate forcing and response and is one of the strongest aspects of the paper.

Finally, the paper ends abruptly without a strong concluding paragraph (perhaps that is the journal format) but I didn’t think the paper fully brought out what the links between Great Basin hydroclimate and forcing tells us about implications for future warming and drying. More detail here would strengthen the manuscript.

The figures are clear and support the manuscript well. Same for the supplementary material.

Specific Comments

Abstract:

Lines 44-45: It is a 736 kyr record, but abstract should acknowledge the ~80 kyr gap from 580 to 660 ka. This becomes important in the discussion where the most robust statistical analysis is really for the younger part of the record.

Lines 48-49: States that the statistical analysis was for the whole record but in the discussion, the analysis appears to pertain to the continuous record from 0 ka to 580 ka. The earliest part of the record does not appear to be part of the analysis. (Lines 244-245 make clear that phase relationships are calculated only over the last 500 kyr.)

Line 56: The implications for future warming and drying are not specified here nor are they really discussed at all in the text. Either remove this sentence, or add some more discussion on the implications of the record for future warming and drying (preferred). This should also be added at the end of the paper.

Body of Manuscript

Paragraph beginning on line 140 and Figure 1: The calcite d18O results show an amplitude of about 2 to 2.5 per mil from glacial maxima to interglacials, whereas the modeling results presented here show amplitudes of no more than 0.6 per mil, and the favored pathway from SNP is closer to 0.5 per mil. These are in the right direction (i.e. more negative for glacials) but these differences in amplitude should be explicitly acknowledged in the discussion. The paragraph gives some explanations as to why the glacials are isotopically lighter citing earlier work – do your model results bear these out? More could be diagnosed from the model to strengthen these interpretations, and it would be helpful to understand why the model underestimates the degree of change. The moisture tagging experiments shown in Figure S5 show the differences in isotopic composition of precipitation from different source regions, but can they also show the effects of enhanced rainout efficiency and cooler STs?

Lines 181- 182:

Lines 317-326: Discussion of controls of d18O includes several processes including temperature dependent rainout of moisture, pathway of storms etc. but doesn't fully draw on the isotope enabled modeling presented earlier in the manuscript. It would strengthen the discussion to use more of the modeling results.

Line 319: Great Basin d18O variability – clarify if you mean the d18O of precipitation or of the calcite.

Line 322: Storm systems move across western North America

Lines 343-344: This section promises insights into periods of rapid warming and drying of the southern Great Basin associated with the last 8 glacial terminations, but only goes into detail for the youngest terminations and has much less detail for TIII to TV, and no discussion of the earlier glacial terminations (one of which occurs during the hiatus so can't be discussed). Either include more discussion of the earlier terminations (preferred) or modify the 8 glacial termination wording.

Lines 343-344: Does age uncertainty preclude discussion of the earlier pre-mid Bruhnes terminations?

Lines 405-406: Refers to Wend et al. 2018 but not the Steidle et al. 2024 paper on water table variations. Seems appropriate to refer to the recently published DH paper.

Lines 409-411: Sentence states that the record provides new constraints on relative P-ET and temperature thresholds, but those are not explicitly stated here. Temperature thresholds for deforestation tipping points would be of broad interest but it is unclear what they are and for that matter, how close we are now to such a tipping point. More detail is needed here.

The paper ends somewhat abruptly without strong concluding remarks about the main findings.

Supplemental material:

Figure S15 shows d13C plotted vs DH and DH2 paleo water table elevations from 1994 and 2018 papers, but not the 2024 paper just published a month ago. Why not? This figure goes back only 350,000 years, but it could be taken all the way back to ~750,000 years in comparison with the Steidle et al. 2024 paper. (This comment goes to the lack of citation of any of the work from the 2024 water table paper).

Version 1:

Reviewer comments:

Reviewer #1

(Remarks to the Author)

Re-review of Controls on the southwest USA hydroclimate over the last seven glacial-interglacial cycles for Nature Communications

I have re-read the paper and the response document from the authors. The authors have done an extremely thorough job in responding to, addressing and expanding upon the ideas/comments from me and the other reviewers. I have no further comments, this works should be published and I anticipate that this will be a highly cited and discussed record for decades. The precise interpretation of the record may change but the secure chronology and robust dataset are essentially unprecedented in terrestrial paleoclimatology.

Reviewer #2

(Remarks to the Author)

I have reviewed the revised version of Wendt et al. and found that the authors have addressed all of my previous concerns. The manuscript has improved considerably and is now close to publication. I only have a few very minor suggestions, none of which should prevent acceptance:

Figure 1: Please consider including a map showing the study area. While I am familiar with the site, many readers, particularly those outside the United States, may not know where it is located.

Line 105: Remove the phrase "the using."

Line 150: Change "...due an early warming..." to "...due to early warming..."

Consistency: The manuscript uses both "paleo" and "palaeo." Please standardize to one form throughout.

Line 172: Revise "...exceptionally wet..." to "...was exceptionally wet..."

Line 238: Revise "...with recent a DH2..." to "...with a recent DH2..."

Line 301: Revise "...linked to mechanisms may influence..." to "...linked to mechanisms that may influence..."

Reviewer #3

(Remarks to the Author)

Review of "Controls on the southwest USA hydroclimate over the last six glacial-interglacial cycles"

Wendt et al.

Resubmitted to Nature Communications

I reviewed an earlier draft of this manuscript and had several suggestions on oxygen isotopes results and interpretation (and using more of the isotope enabled GCM results), using dates and data from the Wendt et al 2024 paper), and the inclusion of a strong concluding paragraph.

After reading through the revised manuscript and the authors' comments to reviewers, I am very pleased with the resulting manuscript. The authors have done a great job addressing the comments of all three reviewers and I think the paper is now acceptable for publication (pending a few very minor comments below).

The revised oxygen isotope change discussion is really strong and the authors then use this to tie isotopes into the statistical analysis to more strongly support interpretations of the climate change mechanisms, especially at glacial terminations. While this apparent before, I feel that the increased discussion of isotopic control mechanisms makes the climatic change interpretations stronger. I also feel that the key points here are improved by removing the millennial-scale mechanisms (per reviewer 2) and focusing on the glacial mid-point analysis.

The new Figure 2 with the DH2 water table data works well and shows the power of combining these proxies and enhances the overall story. The added concluding paragraph does a nice job of bringing the main points of the manuscript home and the paper is greatly improved by adding this.

I don't have any other substantive comments and I really like the way this manuscript has evolved. It is ready for publication with a few minor points below:

Abstract: line 63 – what do you mean by N.H. summer seasonality – does this mean intensity?

Line 65 – "when local recharge reaches < 50% greater than today" is confusing. Would it be better to say "when local recharge declines to <50% of modern"? or something to that effect?

References: Line 1598 or reference 38. Incomplete – if this is a MS thesis, should it have the full title etc.? (I didn't catch this before).

Reviewer 1 suggested some additional references wrt Great Basin lakes that record multiple glacial-interglacial cycles – there is also a new long sediment core record from Stoneman Lake in central AZ (not part of the Great Basin, but relatively close to DH) that records multiple wet glacial - dry interglacial lake alternations. (Staley et al., 2022 GSA Bulletin)

Nature Communications manuscript NCOMMS-24-40790-T

Response to reviewers

The authors thank the Reviewers for their detailed and insightful reviews. Our manuscript has greatly improved as a result. Our point-by-point response to each reviewer is in blue text. Please note that the line # refers to the revised manuscript with “all markup” track changes. The revised manuscript has quite a few track changes – please note that track changes also include unaltered text that has been moved from one portion of the manuscript to another. We also adjusted the spelling of the main text to British English to fit the journal requirements.

Reviewer #1:

Wendt and colleagues present an update to the U-Th dated Devils Hole $\delta^{18}\text{O}$ (and $\delta^{13}\text{C}$) record that includes filling in several important missing gaps and extending the record back significantly beyond 700 kyrs. This site and record is hugely important in understand western US paleohydrology in the Quaternary and this work (and the previous work from this group) is a tour de force. I am not an expert on the spectral analyses conducted, as such I commented primarily on the geochemical, geological and paleoclimate aspects of the work. There are limitations that need to be addressed, but overall I am supportive with moderate revisions of eventual publication of this exciting and important work.

Major comments

The following are major comments that require some addressing but overall nothing here detracts from the overall story and conclusions of the work and I'm sure will be easily handled by the authors:

1) Figure 4 and analysis related to this:

- If Leviathan is going to be used on Figure 4 it should be shown on Figure 3 in my opinion.

Figure 3, now Figure 5, is intended to provide a global context to \$\delta^{18}\text{O}\$ changes recorded in Devils Hole. A comparison of Devils Hole, Leviathan, and Searles lake isotope records are in a separate dedicated plot (Figure S11).

- 239-242 and 255-257: Some sort of definition what Milankovitch and this work considers “near” ice sheets. The southern Great Basin is actually pretty far from the edge of the Laurentide/Cordilleran ice sheet.

We agree with the reviewer that our original text was confusing. This section was reworked and expanded to provide a clearer picture of the mechanism at hand, specifically:

- Line 775: Using the new insights from our iCESM results, potential explanations for this similarity are as follows: (1) fluctuations in greenhouse gas (GHG) concentrations drove regional temperature changes, in which case the DH2 \$\delta^{18}\text{O}\$ would closely follow the atmospheric \$\text{CO}_2\$ record, and (2) changes in the extent of the Laurentide Ice sheet caused a southward displacement of the Pacific Storm Track which influences the amount of depleted southern North Pacific moisture arriving to DH2. In the latter scenario, DH2 \$\delta^{18}\text{O}\$ would closely follow marine records that reflect changes in global ice volume.

What dictates the size of the slice in Figure 4? Is it some sort of implied error bar?

The arc length of the wedge is the phase lag's 95% CI, calculated by cross spectral analysis. We added the following explanation to the caption of Figure 4: 'DH2 \$\delta^{18}\text{O}\$, Leviathan \$\delta^{18}\text{O}\$, global ice volume, and global \$\text{pCO}_2\$ records' phase lag (95% CI) relative to the orbital precession index, in the 22-24 kyr period window. ... Wedges are staggered in height for easier viewing.'

2) Generally speaking I found the references to not be sufficient when I thought carefully about the points the authors were making with respect to the Great Basin lakes. I have made numerous suggestions in this regard below both regarding lake records that do record multiple glacial-interglacial cycles and chronology and compilation efforts over recent years. I hope the authors will consider using more appropriate references for the pluvial lake observations and sediment core based records.

We appreciate this feedback and have added additional references where appropriate.

3) Confounding precipitation $\delta^{18}\text{O}$ as recorded here in groundwater $\delta^{18}\text{O}$ with regional moisture balance changes over glacial-interglacial timescales. As noted in specific comments below, I did not find the writing

clear and all places where broader regional hydroclimate is being interpolated should be scrutinized with respect to phrasing.

We appreciate this feedback, as also mentioned by reviewer 2, and have significantly expanded the sections titled “Controls on orbital-scale $\delta^{18}\text{O}$ variations at Devils Hole 2” to provide clarity. The below paragraph summarizes the interpretation of DH2 $\delta^{18}\text{O}$:

- Line 628: In total, iCESM moisture tagging experiments suggest two key drivers of DH2 $\delta^{18}\text{O}$ variability on glacial-interglacial timescales. First, vapour delivered to DH2 during the LGM is more strongly depleted in $\delta^{18}\text{O}$ for all months due to cooler temperatures and temperate-driven rainout effects (Fig. S4). Second, a change in the proportion of moisture from distinct sources, specifically (i) an increase in depleted moisture from the southern North Pacific due to a southward displacement of the Pacific Storm Track and (ii) a decrease in moisture from the North American continent due to decreased continental recycling during the LGM, as corroborated by proxy data (54). iCESM does not fully resolve the NAM; we therefore cannot rule out NAM-related processes as potential contributors to DH2 $\delta^{18}\text{O}$ on glacial-interglacial timescales. Simulated change in $\delta^{18}\text{O}$ between LGM and PI ($\Delta\delta^{18}\text{O}$) underestimates the observed $\Delta\delta^{18}\text{O}$ in DH2 record by $\sim 1\text{‰}$ (considering seawater corrections). This may be due to (1) limitations in iCESM to simulate processes related to NAM strength, (2) lower LGM-PI temperature differences in iCESM simulations ($\Delta 5^\circ\text{C}$) relative to the truly magnitude suggested by proxy reconstructions, and/or (3) inaccuracies in iCESM ice volume forcing in the southern Sierra Nevada mountain range (62, 63), which may alter moisture trajectories and contribute to increased rainout during glacial periods.

4) Groundwater residence times feeding the DH/DH2 record. The authors acknowledge the residence time being $\sim 800+$ years early on in the paper, but I was surprised that:

- This was not accounted for in the lead-lags that were analyzed, especially given that these leads and lags would be variable given the recent groundwater modeling by Jackson et al. (2023) (a paper which several of the authors here are also on).

We agree that that groundwater transit times are an uncertainty that was not accounted for in our original text. We have (i) updated Figure 4, (ii) updated Figure S12, and (iii) re-calculated the lead-lag analysis and cross-spectral analysis both with and without groundwater residence times, as explained below:

- Line 787: For each analysis, we examined the phasing of DH2 $\delta^{18}\text{O}$ with and without a 880-year groundwater residence time (i.e. adding 880 years to the DH2 $\delta^{18}\text{O}$ chronology over the last 500 kyrs), which is the estimated modern residence time from recharge centres to DH and DH2 caves (21, 23, 24). Modern groundwater transit times are considered a conservative maximum, as transit times were likely shorter during glacial periods due to the significantly higher ($>250\%$) at the LGM) recharge to the local aquifer (25).

Figure 4. Devils Hole 2 (DH2) $\delta^{18}\text{O}$ power spectrum and phasing analysis. A: DH2 $\delta^{18}\text{O}$ power spectrum of 100 OxCal age models. Each power spectrum has been normalized to its corresponding 95% false alarm record, such that power >1 is interpreted as a frequency significant above red noise. The black line is the

mean normalized power of the 100 records calculated at each frequency. The blue circles highlight local maximum mean normalized power above 1, with 2σ error bars from the 100 records. The red dash dot line represents the theoretical red noise (first-order autoregressive process). B: DH2 $\delta^{18}\text{O}$, Leviathan $\delta^{18}\text{O}$, global ice volume, and global pCO_2 records' phase lag (95% CI) relative to the orbital precession index, in the 22-24 kyr period window. Zero phase (pointing up) is set as precession index minimum, equivalent to the Northern Hemisphere summer solstice (June 21st) insolation maximum, and arrows mark direction of increasing years of lag from the set zero phase (1 cycle = 23 kyr). Wedges are staggered in height for easier viewing. Datasets: DH2 $\delta^{18}\text{O}$ on its mean age model (this study) and adjusted 880 years to account for maximum groundwater recharge time (this study, blue), Leviathan $\delta^{18}\text{O}$ from Nevada stalagmites (13), atmospheric CO_2 concentrations (pCO_2) on the AICC2023 chronology (56, 57) and global ice volume inferred from the absolute-dated Red Sea RSL (58).

Figure S12: Probability density curves fitted by the Gaussian distribution of several climate records (see Table S5) and their respective phasing relative to DH2 $\delta^{18}\text{O}$. Negative values indicate that DH2 $\delta^{18}\text{O}$ leads, positive values indicate that DH2 $\delta^{18}\text{O}$ lags. Right panel: as in left, but including an 880-year groundwater transit time (i.e. adding 880 years) to DH2 $\delta^{18}\text{O}$.

- Wendt et al. reference 74 and the observed variations in $(^{234}\text{U}/^{238}\text{U})$ are significant in that the variations perhaps even allow us to understand the groundwater paleohydrology from the DH cores. The authors did not mention or comment on this whatsoever despite this clearly being relevant.

We agree and have added the following to line 431:

- The magnitude of DH2 $\delta^{18}\text{O}$ is also negatively correlated with DH2 $\delta^{234}\text{U}_i$; maxima during each glacial period (35). DH2 $\delta^{234}\text{U}_i$ is interpreted as a proxy for water-rock interactions associated with a fluctuating water table, with high values corresponding to periods of high groundwater recharge (35). For example, exceptionally low $\delta^{18}\text{O}$ values and relatively high $\delta^{234}\text{U}_i$ maxima (1850‰) during Marine Isotope Stage (MIS) 10 suggest that this glacial period exceptionally wet and cool in the southern Great Basin.

Minor Line-by-Line Comments

48: "Full-record analysis" – this is too much jargon. I realize the need for brevity in this journal's format but please clarify this sensen.

We have substituted full-record analysis for lead-lag analysis.

65: The references 1-4 included here other than Lowenstein et al. 1999 are not suitable. They only cover the last glacial and deglaciation. I would suggest instead citing work from Owens Valley (from the >800 kyr OL-92 core), Searles (recent work by Peapple/Stroup and older work by Lowenstein), Manix (Reheis et al., 2012) and Bonneville/Bear Lake (cores that go back 2-3 glacial cycles) that shows “repeated expansion” of the lakes. I understand that the references used here are somewhat canonical in the pluvial lake literature but to make the point being made in this sentence of repeated pluvial lakes over glacial-interglacial cycles some other references are needed. You might also consider citing Reheis et al. (2014) and the recent paper by Lowenstein et al. (2024) on Death Valley.

We appreciate the suggestions. Lowenstein et al. (2024), Peapple et al. (2022), Reheis et al. (2012), and Bischoff et al. (1997) were added. References that only extend over the last glacial period were removed.

69: Remove “the” before “Milankovitch”
Adjusted.

84-86: these references, 12-15, are good references but largely only acknowledge circulation change. But the sentence is focused on the expansion of pluvial lakes and the attribution of temperature (via evaporation) vs moisture delivery. I suggest you cite work by McGee et al. (2018) and Ibarra et al. (2018) here as the role of evaporation on the expansion of pluvial lakes is also non-trivial.

We agree and have made two changes. First, this portion of the introduction was moved to the “Controls on orbital-scale $\delta^{18}\text{O}$ variations at Devils Hole 2” section to allow space for further explanation. Second, we added the following text to line 345 and reference Ibarra et al., 2018. We chose not to reference McGee et al. (2018) as it related to millennial-scale mechanisms, but instead cited Tabor et al. (2021).

- Line 496: Wetter glacial conditions are attributed to cooler temperatures and suppressed evaporation (40, 41) coupled with a southward displacement of the Pacific Storm Track (40, 42-44) that resulted in a drier northwest United States and wetter southwest (opposite to modern day).

83-93: The rest of this paragraph is largely about precipitation $\delta^{18}\text{O}$ (or dD) rather than hydroclimate (P-E and/or temp vs. precipitation) \diamond overall I think the authors could do a much better job defining what they mean by “hydroclimate change” – do they merely mean the $\delta^{18}\text{O}$ (or dD) of precipitation? Rather than effective moisture and/or regional water balance and/or moisture divergence and convergence (as implied by some of the papers that are cited in this paragraph)? More precise writing is needed here.

We agree with this feedback and have moved this portion of the introduction to the “Controls on orbital-scale $\delta^{18}\text{O}$ variations at Devils Hole 2” section for further explanation. Throughout the paper we substituted the term “hydroclimate” for more specific terms and descriptions, including the introduction and conclusion.

99: Confusing that then in the next paragraph it is plural (since DH and DH2 are separate caves but the same groundwater system) – it would be good to be consistent/clarify.

Agreed, we now distinguish between the two caves throughout the paper.

140-145: I don’t disagree with this overall conclusion at the broad scale, but I will mention that the authors are brushing over a lot of work done using modern, higher resolution back-trajectory analysis from nearby that also shows the impact of the high southern Sierras, that presumably would be amplified by the Sierra Nevada ice cap leading to wrap around effects and increased moisture rainout during glacials. See the back trajectory analyses by Lechler and Galewsky (2013, Geology), Mix et al. (2019, EPSL) and Gagnon et al. (2023, GSAB). My suggestion is to expand on the sentence from line 145-147 and acknowledge this issue.

Great point. The iCESM simulations we examine cannot resolve the Sierra Nevada ice cap at the LGM. It’s possible this process contributed to the discrepancy in change in $d^{18}\text{O}$ between model and proxy. We added the following text to line 614 and cited the reviewer’s suggested references:

- Simulated change in $\delta^{18}\text{O}$ between LGM and PI ($\Delta\delta^{18}\text{O}$) underestimates the observed $\Delta\delta^{18}\text{O}$ in DH2 record by $\sim 1\text{‰}$ (considering seawater corrections). This may be due to (1) limitations in iCESM to simulate processes related to NAM strength, (2) lower LGM-PI temperature differences

in iCESM simulations ($\Delta 5^\circ\text{C}$) relative to the truly magnitude suggested by proxy reconstructions, and/or (3) inaccuracies in iCESM ice volume forcing in the southern Sierra Nevada mountain range (62, 63), which may alter moisture trajectories and contribute to increased rainout during glacial periods.

165-166: How much excess ^{230}Th is there in core D in the Holocene or at that depth today? Would be best to be specific rather than just “not significantly affected”

This issue is discussed in depth in Moseley et al. 2016 *Science*, which is referenced.

194-195: Note that Bhattacharya et al. (2024, Elsevier) commented on this and the apparent magnitude of δD vs $\delta^{18}\text{O}$ variations (accounting for meteoric water line slopes).

Thanks for pointing this paper out.

282-284: Add refs for these records that were used.

Done.

305-307: Have the authors considered generating a LGM to Holocene record from one of the other cores near D to get around this problem? (I realize this is outside the scope of this paper/study).

Although beyond the scope of this immediate study, there is ongoing work with additional cores near core D.

325: define fast? Also what about the 900 year water residence time of the system?

The word “fast” was removed.

336-338: Fix reference formatting, the full list of authors is not needed here.

Fixed.

352-353: With all due respect to the author of reference 65, the geochronology and compilation literature from the last decade or so of paleo lake high stand datasets should be cited as that is the actual geological data. References such as - Munroe and Laabs (2013); Ibarra et al. (2014); Reheis et al. (2014); McGee et al. (2018); Hudson et al. (2019) - rather than a modeling paper. I do think somewhere reference 65 should be included, perhaps when other similar work by Oster or others is cited.

The in-depth discussion on millennial-scale fluctuations in paleo lake levels has been deleted.

353-356: Similar to my comments in the introduction the authors seem to be confounding $\delta^{18}\text{O}$ variations with P-E/effective moisture in these sentences.

Fixed, as detailed above.

358-360: Isn't this entirely expected given the 800+ year residence time of water in the system?

Yes, the in-depth discussion on millennial-scale fluctuations in paleo lake levels has been deleted.

380-400: What is the residence time of DIC in the groundwater system relative to the water itself? Is this known from radiocarbon measurements for example or not possible due to carbonate bedrock contributions to DIC?

Great question. ^{14}C concentrations in DIC along the flow path of the modern Ash Meadows ground-water basin suggests extensive carbon exchange with the bedrock (Copen et al., 1994 *Science*). DIC ^{14}C travel-time calculations must therefore be corrected for mineral/gas dissolution, mineral/gas precipitation/exsolution, cation exchange, and carbon isotopic exchange—all of which can significantly alter the amount of DIC ^{14}C in groundwater.

Instead, Thomas et al. 2021 *Applied Geochem.* used DOC ^{14}C to calculate a travel time from Spring Mountains to Devils Hole cave of ~2900 years. They conclude that carbon residence times are slightly longer but similar to groundwater residence times calculated from hydrogeologic data (880 years). We have added this point to line 1435:

- Dissolved inorganic carbon (DIC) residence times cannot be reliably calculated in the DH2 aquifer due to extensive carbon exchange between groundwater and carbonate bedrock (23, 69) while

dissolved organic carbon (DOC) $\delta^{13}\text{C}$ measurements suggest that DOC transit times are similar to but longer than groundwater transit (23). It is therefore surprising that the $\delta^{13}\text{C}$ signal arrives before $\delta^{18}\text{O}$, as first pointed out by Coplen et al. (1994), and suggest alternative forcings from $\delta^{18}\text{O}$ of precipitation.

410-411: This phrasing is a bit over the top.
Agreed, sentence deleted.

411: The paper just ends – this is a bizarre ending to a very worthwhile study. While the format/word limit must be coming into play here the reader is left with no summary or higher-order connection being made and (bizarrely) is left on a discussion of the $\delta^{13}\text{C}$ record that was not mentioned until the last few paragraphs.

We agree. This was an artifact of the strict word count of our original submission to Nature Geoscience. We have now added the following conclusion paragraph:

- In summary, the phasing of DH2 $\delta^{18}\text{O}$ timeseries suggests that temperature-related processes are dominant drivers of orbital scale $\delta^{18}\text{O}$ variability in southern Great Basin precipitation, consistent with iCESM model outputs. Global ice volume lags DH2 $\delta^{18}\text{O}$ on average, suggesting that mechanisms linked to the North American ice sheets (e.g, shifting storm track) contributed to, but were not the primary driver of, DH2 $\delta^{18}\text{O}$ variability. In contrast, the DH2 $\delta^{13}\text{C}$ timeseries is in-phase with seasonal insolation during the warm growing season (between summer solstice and fall equinox). Prominent $\delta^{13}\text{C}$ lows, indicating high primary productivity, coincide with peak boreal summer insolation during the last six interglacial periods. A rapid decrease in vegetation density coincides with warm interglacial temperatures and <50% greater recharge relative to today, indicating a past tipping point for environmental decline. This study sheds new light on the relationship between temperature, moisture balance, and vegetation in the southern Great Basin on orbital timescales. Our results underscore the link between increased CO_2 concentrations and regional warming (17, 18, 71, 72), which is expected to contribute to reduced effective moisture in the Great Basin over the coming century (15, 19).

446: I was very surprised not to see some discussion of the Wendt et al. ($^{23}\text{U}/^{238}\text{U}$) variation observations relative to the groundwater table and newly compiled $\delta^{18}\text{O}$ record.

We agree and have added the following to line 431:

- The magnitude of DH2 $\delta^{18}\text{O}$ is also negatively correlated with DH2 $\delta^{234}\text{U}_i$ maxima during each glacial period (35). DH2 $\delta^{234}\text{U}_i$ is interpreted as a proxy for water-rock interactions associated with a fluctuating water table, with high values corresponding to periods of high groundwater recharge (35). For example, exceptionally low $\delta^{18}\text{O}$ values and relatively high $\delta^{234}\text{U}_i$ maxima (1850‰) during Marine Isotope Stage (MIS) 10 suggest that this glacial period exceptionally wet and cool in the southern Great Basin.

546-550: $\delta^{18}\text{O}$ not formatted correctly.
Fixed.

Please note that some of the supplementary figures are not uniformly formatted - I would suggest this be fixed in revision.
Fixed where appropriate.

Reviewer #2:

The manuscript by Wendt et al. analyzed data from Devils Hole in the Great Basin to investigate a ~700ky hydroclimate record. The study shows that the Great Basin's drylands experienced significant hydroclimate shifts throughout the Quaternary and that these shifts are linked to large-scale climate changes in the eastern North Pacific and western North America. The study also shows that Devils Hole $\delta^{18}\text{O}$ correlates with variations in atmospheric CO_2 levels and Northern Hemisphere ice volume across seven glacial-interglacial cycles. Particularly, increases in $\delta^{18}\text{O}$ during glacial terminations were associated with rising CO_2 , potentially indicating a rapid climate response to temperature changes and

radiative forcing. The manuscript is well-written, and the results are very interesting and well-detailed, which allowed me (a non-expert on Great Basin hydroclimate and spectral analysis) to understand the work and its implications. I have few major and minor comments that I hope are found useful by the authors.

Major comments:

Models shown in Figure 1B suggest that northern North Pacific sources of moisture are going to be depleted compared to southern North Pacific sources. If during the LGM southern North Pacific sources of moisture became more prominent due to a southern displacement of the jet, wouldn't that result in more positive $\delta^{18}\text{O}$?

We agree that this question was not sufficiently addressed in the original manuscript and thank reviewer 2 for highlighting this potential point of confusion. We have edited Figure 1b (now Figure 3b) to include the LGM and PI $\delta^{18}\text{O}$ values of southern vs northern Pacific waters and added the following sentence to line 542:

- Despite its lower latitude source, southern North Pacific water vapour arrives to DH2 ~2‰ more depleted relative to northern North Pacific vapour (Fig. 3) likely due to longer moisture trajectory paths and/or higher rainout efficiency resulting from a greater land-sea temperature gradient. The proportion of precipitation sourced from the North American continent decreased during the LGM (Fig. 3) due to suppressed re-evaporation from land sources as a result of cooler terrestrial surface temperatures.

Figure 2: Modern moisture trajectories and iCESM results. (A) Cluster analyses of modern rain bearing trajectories arriving at Devils Hole caves from September 2007 to August 2011 (36). Great Basin is outlined in white. Colors are used to distinguish clusters, with the width of each cluster representative of the number of trajectories. (B) Modeled Last Glacial Maximum (LGM; blue) versus pre-industrial (PI; orange) change in source % and $\delta^{18}\text{O}$ of precipitation at Devils Hole caves sourced from the North American continent (NA), Northern North Pacific (NNP), and Southern North Pacific (SNP). All other minor moisture sources (<5% source in PI) shown in Fig. S5. (C) Modeled LGM versus PI change in precipitation over western North America. Devils Hole caves indicated by white star.

Lines 258-261: The authors argue that there is not a “simple 1:1 link between North American ice volume and DH2 $\delta^{18}O$ on orbital timescales, but instead for changing boundary conditions and/or multiple feedback processes that act in tandem to vary the lag times of DH2 $\delta^{18}O$ relative to maximum summer insolation.” My understanding is that ice volumes also lag summer insolation by approximately 5,000 years and that ice sheet response is a result of the complex interplay between the Earth’s orbital variations, the thermal inertia of ice sheets, and the dynamics of the global climate system. This is in some ways shown in Figure 4 and highlights that both ice sheet volume (based on SL curves) and DH2 $\delta^{18}O$ are similar, or at least within age model uncertainty. It is my opinion that, even though a simple link cannot be established, there is no way to statistically separate them. In the end, it is a very complicated system, involving ice dynamics, atmospheric circulation changes, and water recharge lag times that probably complicate the correlation of the $\delta^{18}O$ record with other time series.

We agree that the original text lacked a more thorough discussion of the complexity of forcings. We have altered the following text:

- Line 1008: Next, we performed the same cross spectral analysis on an absolute-dated relative sea level (RSL) curve that reflects changes in global ice volume (58) and atmospheric CO₂ concentrations (56, 57). (Fig. 3). The records’ 95% CI phase lag range relative to orbital forcing underscores the complex interplay between Earth’s orbital variations and ice sheet growth and atmospheric CO₂ (63). The multi-thousand-year phase lag of maximum DH2 $\delta^{18}O$, ice volume, and atmospheric CO₂ to summer insolation maximum in the precessional phase overlaps between the three records (Fig. 4). This overlap is not unique to DH2, as shown by the Nevada stalagmite $\delta^{18}O$ composite (Leviathan record) (12, 13) in Fig. 4. We therefore cannot rule out North American ice volume (through the mechanisms outlined above) nor temperature changes (through GHG radiative forcing) as potential drivers of southern Great Basin $\delta^{18}O$ variability at orbital timescales.
- Line 1376: Overall, our phasing and midpoint analyses suggest that DH2 $\delta^{18}O$ bears the closest structural and temporal similarity to atmospheric CO₂ variability, both when calculating an average over the entire record and over short periods of abrupt warming and drying. In the event of a termination, rising CO₂ concentrations would warm eastern North Pacific and western North America regions which, in alignment with our iCESM results, would (1) enrich the $\delta^{18}O$ of water vapor from moisture sources, (2) decrease land-sea temperature gradients (decreasing rainout efficiency along moisture trajectories), and (3) increase evaporation from the North American continent, collectively resulting in an enrichment of $\delta^{18}O$ moisture arriving to DH2. Phasing using the lagged correlation method and midpoint analysis shows that, on average, DH2 $\delta^{18}O$ leads changes in global ice volume. This suggests that the modulation of the Pacific Storm Track by North American ice sheets may have contributed to (e.g., amplified), but was not a primary driver of, changes in DH2 $\delta^{18}O$ on orbital timescales. Another example of the close coupling between DH2 $\delta^{18}O$ and CO₂ is at the onset of terminations. The timing of DH2 $\delta^{18}O$ rise closely aligns with the onset of CO₂ rise associated with TII-V, including the “early” rise in DH2 $\delta^{18}O$ and CO₂ at the onset of Termination IV (358-345 ka) which is absent from global ice volume records (Fig. 5).

A main concern is the relationship established between Great Basin hydroclimate and atmospheric CO₂ concentrations. The authors argue for this relationship by analyzing the records in two ways and show that $\delta^{18}O$ and CO₂ are strongly correlated, but correlation does not always imply causation. The authors state “While continental ice sheets remain a major control of landscape moisture availability in western North America (12-15), which has been shown to partially influence Great Basin $\delta^{18}O$ (15), the phasing of DH2 $\delta^{18}O$ also suggests a fast-acting climate response to regional temperature changes driven by greenhouse gas concentrations.” It is my opinion that there is a strong correlation between all the proxy records analyzed. Authors should not rule other correlations as potential hydroclimate forcings out just because CO₂ and $\delta^{18}O$ have a slightly stronger correlation.

The text has been altered to emphasize that we do not rule out potential forcings (as discussed above), but that lead-lag analyses clearly show DH2 $\delta^{18}O$ leading global ice volume. As such, our argument that global ice volume is not a primary driver of DH2 $\delta^{18}O$ remains.

- Line 1392: In total, results from two of the three approaches used here suggest that temperature changes are the primary driver of DH2 $\delta^{18}O$ variability on orbital timescales, with secondary

drivers stemming from atmospheric circulation changes due to North American ice sheet extent. Both mechanisms outlined here are consistent with orbital forcings (i.e. Milankovitch theory). Changing boundary conditions may amplify or dampen the relative influence of a particular driver on DH2 $\delta^{18}\text{O}$ over time, while variable groundwater transit times may vary relative leads and lags.

The entire section of the manuscript related to “Terminations in the Great Basin” is confusing and vague. The authors argue that there is a link between the maximum rate of change in $\delta^{18}\text{O}$ and speleothem records from China. However, the synoptic scale changes suggested to explain the correlation between the East Asian Monsoon and Great Basin hydroclimate are vague. Why is the maximum rate of change used? What does this rate of change represent? The authors state that the “link between the East Asian Monsoon and Great Basin hydroclimate may be as follows: meltwater entering the North Atlantic alters ocean and atmospheric circulation and associated heat fluxes, prompting an enrichment of the East Asian monsoon water isotope (62, 64). Around this time, paleo lake high stands are recorded across the southern Great Basin (65).” but then fail to explain why meltwater entering the North Atlantic Ocean could result in a wetter climate in the western North America. The authors then state “The mechanisms that drive millennial-scale wet events in the Great Basin are debated (66-68), with recent studies suggesting increased delivery of Pacific moisture due to a weakening of the North Pacific Hadley circulation in response to northern high-latitude cooling (69, 70).” but this is now talking about atmospheric circulation in the Pacific Ocean, unrelated to North Atlantic meltwater.

We agree that this section was confusing and dove too deep the weeds of millennial-scale variability. As a result, the main point of our termination midpoint analysis (to determine the lead and lags or various paleo records at terminations) is lost. Rather than expanding the explanation of millennial-scale mechanisms, **we have deleted this section entirely and moved the midpoint analysis to the “Orbital-scale controls on the Great Basin hydroclimate.”**

- Line 1358: Lastly, we investigated potential drivers of DH2 $\delta^{18}\text{O}$ during periods of abrupt warming and drying. To do so, we determined the midpoint of TI-V in DH2 $\delta^{18}\text{O}$ and the aforementioned records on their individual chronologies. The midpoint of the ascending limb of DH2 $\delta^{18}\text{O}$ during TII (132.15 ± 1.5 ka) was previously calculated by (10). Using the same approach, we calculate the midpoint associated with TIII (244.0 ± 1.1 ka), TIV (341 ± 3 ka), TV (430 ± 6 ka), TVI (529 ± 6 ka) and TVIII (708 ± 8 ka) of the DH2 $\delta^{18}\text{O}$ record (table S5; see methods). Maximum groundwater residence time (880 yrs) falls within chronological uncertainties of each midpoint. As shown in (10), multiple growth rate changes in core D during TI and the Holocene resulted in poor age control (up to 18% relative 2σ uncertainty between 18-10ka). For this study, we focus on TII-TV during which the DH2 record has the best age control ($\leq 1\%$ relative 2σ uncertainties) (Fig. 6). The termination midpoints of DH2 $\delta^{18}\text{O}$ and selected climate records are listed in table S5. The greatest agreement in the timing of TII-TV midpoints occurs between DH2 $\delta^{18}\text{O}$ and atmospheric CO_2 on the AICSTAL2024 chronology (Fig. 6). DH2 $\delta^{18}\text{O}$ midpoints lagged CO_2 by an average of ~ 100 years, ranging from -0.5 to $+0.5$ kyr, where negative values indicate that DH2 $\delta^{18}\text{O}$ lags CO_2 . Following atmospheric CO_2 , the absolute-dated RSL curve shows variable leads and lags of termination midpoints, with RSL rise lagging DH2 $\delta^{18}\text{O}$ by an average of 2.7 kyr (-0.4 to 7.2 kyr, where negative values indicate that DH2 $\delta^{18}\text{O}$ lags RSL) during TII-V.

Minor comments:

Line 81: This part is a bit confusing to me as the new DH record is called DH2 but DH2 is also Devils Hole 2.

Agreed, fixed.

Line 89-90: Can you please specify what are the challenges in the interpretation of $\delta^{18}\text{O}$?

Yes, we now expand the current debate over the interpretation of $\delta^{18}\text{O}$ starting on line 540:

- The exact mechanisms that drive DH2 $\delta^{18}\text{O}$ depletion during glacial periods remain unclear. Previous studies (6, 10) suggest that glacial-interglacial DH2 $\delta^{18}\text{O}$ variations are partially driven

by changes in the proportion of summer precipitation sourced from NAM (high $\delta^{18}\text{O}$) to winter precipitation sourced from the Pacific (low $\delta^{18}\text{O}$) at DH2. As a result, low $\delta^{18}\text{O}$ values during glacial periods are partially due to a larger proportion of low- $\delta^{18}\text{O}$ cool-season rainfall associated with (i) a southerly displaced Pacific Storm Track and (ii) a weakened NAM. Another proposed mechanism is temperature: terrestrial proxies suggest that the southern Great Basin was 6-10°C cooler during the last glacial maximum (LGM) relative to preindustrial (3, 40, 46, 47) and SSTs reconstructed from the moisture source regions of DH2 were 2-5°C cooler (48, 49). Tabor et al. (2021) suggest that a greater land-sea temperature gradient during the LGM increased the rainout efficiency of moisture trajectories moving inland to the Great Basin. This effect, when coupled with cooler LGM temperatures that suppressed evaporation, would result in lower $\delta^{18}\text{O}$ of precipitation during glacial periods (40, 50). Groundwater temperatures in the DH2 aquifer have remained constant ($\pm 1^\circ\text{C}$) over the last 500 ka (22, 51) and thus have a negligible effect on $\delta^{18}\text{O}$ variations in DH2 calcite.

Line 92-93: I find this comment vague. Based on your results, can you add what are the implications of higher atmospheric CO₂ on Great Basin hydroclimate? Would we expect an even drier/warmer climate? Agreed, this line has been edited to:

- Understanding these mechanisms has become increasingly urgent, as warmer temperatures over the next century are expected to reduce water availability in this already water-scarce yet increasingly populated region (15-19). And two sentences in the concluding paragraph were added:
- Line 1609: In total, the extended DH2 $\delta^{13}\text{C}$ record suggests that seasonality is the dominant driver of orbital-scale environmental change in the highlands of southwest Nevada, with a tipping point in effective moisture (<50% greater recharge relative to today) that result in a rapid and unilateral decline in primary productivity during warm interglacials.

Line 320: Change "...DH2 $\delta^{18}\text{O}$ and atmospheric CO₂ consistent with the moisture..." to "...DH2 $\delta^{18}\text{O}$ and atmospheric CO₂ is consistent with the moisture..."

Line 322: Change "...storm systems move western North America." to "...storm systems move towards western North America." or "...storm systems move into western North America." Not sure what works here but there is something missing in the original sentence.

Line 325: Change "...phasing of DH2 $\delta^{18}\text{O}$ also suggests a..." to "...phasing of DH2 $\delta^{18}\text{O}$ with CO₂ also suggests a..."

The paragraph that includes lines 320-325 in the original manuscript has been changed to the following (now on line 1348):

- Overall, our phasing and midpoint analyses suggest that DH2 $\delta^{18}\text{O}$ bears the closest structural and temporal similarity to atmospheric CO₂ variability, both when calculating an average over the entire record and over short periods of abrupt warming and drying. In the event of a termination, rising CO₂ concentrations would warm eastern North Pacific and western North America regions which, in alignment with our iCESM results, would (1) enrich the $\delta^{18}\text{O}$ of water vapor from moisture sources, (2) decrease land-sea temperature gradients (decreasing rainout efficiency along moisture trajectories), and (3) increase evaporation from the North American continent, collectively resulting in an enrichment of $\delta^{18}\text{O}$ moisture arriving to DH2. Phasing using the lagged correlation method and midpoint analysis shows that, on average, DH2 $\delta^{18}\text{O}$ leads changes in global ice volume. This suggests that the modulation of the Pacific Storm Track by North American ice sheets may have contributed to (e.g., amplified), but was not a primary driver of, changes in DH2 $\delta^{18}\text{O}$ on orbital timescales. Another example of the close coupling between DH2 $\delta^{18}\text{O}$ and CO₂ is at the onset of terminations. The timing of DH2 $\delta^{18}\text{O}$ rise closely aligns with the onset of CO₂ rise associated with TII-V, including the "early" rise in DH2 $\delta^{18}\text{O}$ and CO₂ at the onset of Termination IV (358-345 ka) which is absent from global ice volume records (Fig. 5).

Line 366: Change "...with an acceleration Northern Hemisphere..." to "...with an acceleration of Northern Hemisphere..."

This line has been deleted.

Figure 1: The width of the trajectories has to be better explained. I understand that it represents clustered trajectories. Is it a percentage of trajectories? Is it just a relative number between each other? There is also reference in the manuscript to particular storm tracks, such as the Pacific storm track. Do all these tracks have names? If so, they should all be added to the figure.

Figure 1. LGM and PI should be defined within the figure caption.
Fixed.

Figure 3: Y axis of Devils Hole $\delta^{18}\text{O}$ record should be “Devils Hole 2”.
Fixed.

Figure 5: The dashed line for the CO₂ curve mid-point is not shown.
The timing of midpoint is so similar that the dashed lines overlap.

Figure S3: There is something wrong with the embedded text in this figure.
We are unsure what the reviewer is referring to. We will save the supplementary material as a .pdf to avoid weird formatting issues that commonly arise with Word.

Figure S9: No tie points are shown. Y-axis delta symbols need to be changed.
We are once again unsure what the reviewer is referring to, as the vertical lines (indicating tie points) are clearly shown. Similar to the δ symbols. As per above, perhaps the .doc was corrupted.

Figure S13: There is something wrong with part b. Why do you have months instead of phase lag, such as that shown in part a? The manuscript text discusses lags in obliquity as well, but this is not shown in Figure S13. In lines 387-388, Figure S13b is explained but I think there is something off here. Would it be called differently than $\delta^{13}\text{C}$ vs precession index phasing?

Thank you for pointing out the confusion cause by original Figure S13. We have expanded the figure caption to better explain the figure subplots a and b: ‘Phase lag relative to the orbital precession index for DH2 $\delta^{18}\text{O}$ (a) and $\delta^{13}\text{C}$ (b) (DH2 record on the mean age model), shown on phase wheels. Zero phase (pointing up) is set as alignment with precession index minimum, equivalent to the Northern Hemisphere summer solstice (June 21st) insolation maximum. Note, the Gregorian calendar date of June 21st for boreal summer solstice is approximate due to variable season length set by the precession index. In (a), phase wheel tick marks note years of lag from precessional index minimum (in a 23,000 year cycle, a 1/4th cycle lag is equivalent to 5,750 years lag). Arrows mark the direction of increasing years of lag from the set zero phase. In (b), phase wheel tick marks note alignment with other points within the precessional cycle: pointing up is alignment with boreal summer solstice, while pointing to the right is alignment with boreal fall equinox. A record’s alignment with boreal fall equinox is equivalent to 5,750 years lag behind boreal summer solstice in the precessional index.’

The figure is designed to assist the main text, which discusses phase lag from summer solstice for the $\delta^{18}\text{O}$ record, and alignment with the summer growing season (between summer solstice and fall equinox) for the $\delta^{13}\text{C}$ record. Figure S13 is used to show lead/lag in the precessional index. Figure S14b has been added to demonstrate DH2 $\delta^{18}\text{O}$ lag with obliquity (DH2 $\delta^{13}\text{C}$ does not have a significant spectral peak at 40 kyr period).

Reviewer #3 (Remarks to the Author):

Controls on the southwest USA hydroclimate over the last seven glacial-interglacial cycles

Recommendation: Publish after moderate revision

This manuscript presents one of the longest absolute-dated, terrestrial paleoclimatic records from Devil’s Hole, Nevada which extends the original DH record back to 736 ka (the new record is called DH2). The primary proxy is a high-resolution set of oxygen isotope measurements from the calcite and while most of the U-series dates constraining this record (114 in total) were previously published, this paper provides one new ²³⁰Th-²³⁴U date and three new ²³⁴U-²³⁸U dates for the older part of the record. The two

calcite vein cores comprising the record are taken near the water table in Devils Hole 2, eliminating the offset in U-series ages relative to water depth that provided anomalously young ages in the original DH record. While the oldest part of the record dates to 736 ka, there is a depositional gap from ~580 ka to 660 ka which precludes analysis / interpretation of climatic changes over this interval. The older part of the record is time constrained by the new 234U-238U dating technique that relies on understanding the past $d^{234}\text{U}$ initial in the DH calcite (discussed in Li et al., 2021).

The results presented here build on those of Moseley et al., 2016, showing that hydroclimatic changes in the southern Great Basin are driven by orbital-scale processes back to 736 ka (with the age gap caveat noted above), and they build on the original DH record with a new, higher-resolution set of $d^{18}\text{O}$ measurements. Using an isotope-enabled GCM, the DH2 record is interpreted in terms of large-scale changes across North America (ice sheets, temperature and evaporation) and Pacific SSTs. Specifically, the southern Great Basin is interpreted to have received higher winter precipitation amounts with that moisture sourced from the southern North Pacific and carried along a southerly displaced storm track.

The long-term record presented here will be of broad interest to the paleoclimate community and the modeling results interpreted in tandem with the oxygen isotope data help further constrain the climate dynamics associated with the dramatic changes in hydroclimate over multiple glacial-interglacial cycles. The detailed statistical analysis that a well-dated record like this allows shows the importance of rising CO_2 levels at glacial terminations as being a key driver of local climate change.

The paper is well-written and the conclusions are important enough to merit publication in Nature Communications. However, before the paper can be published, there are a few minor to moderate revisions, that include citation of another recently published DH2 record by the same group of authors, more detail on the interpretation of the climate model results and further and more detailed discussion of the implications of this work.

First – a significant paper detailing water table fluctuations in Devils Hole 2 was published online July 14, 2024 in Communications earth & environment (“Moisture availability and groundwater recharge paced by orbital forcing over the past 750,000 years in the southwestern USA) by the same group of authors as this manuscript. This paper is not cited nor referred to at all in the current manuscript despite being from the same location and covering roughly the same time interval (750 kyr vs. 736 kyr). I assume the reason for this is that Nature Communications does not allow citations of in press work until it is published. I’ll also note that the proxy records are different – the published paper records water table variations in Devils Hole 2 from a different set of calcite cores (cores M, R, L, K, H, I, all of which are taken from above the current water table in DH2) than the current manuscript (core D which is the primary record with spliced in intervals from core P). Each set of cores has its own unique set of radiometric dates, and no stable isotope work was done in the published paper, thus these are different proxies.

However, this manuscript would benefit from including the results of this work as it covers the same time interval (and in fact includes the 80 kyr gap from the current work), and it supports the orbital scale delivery of more precipitation during glacial periods. Figure S15 shows results from this paper plotted against earlier, shorter records of DH water table fluctuations – back to about 350 ka – so including this most recent work would make this correlation much stronger. I assume now that this other work is fully published, that it can be cited and used to support the current manuscript.

Yes, the reviewer is correct. At the time we submitted the 750-ka water table paper was still in revision, and thus could not be cited (per the rules at Nature). We now reference it several times in the paper and added a new figure (Figure 2), as listed below:

- Line 426: Palaeo water table reconstructions from DH (34) and DH2 (27, 28) caves reflect past changes in groundwater recharge amount to the local aquifer over the last 750 ka (25, 27) (Fig. 2). At TII, the rise in DH2 $\delta^{18}\text{O}$ towards interglacial values coincides with a multi-metre drop in the local palaeo water table (10, 27) and the desiccation of pluvial lakes in Death Valley and Searles Valley in the southern Great Basin (onset of drying at $137.6 \text{ ka} \pm 0.5 \text{ ka}$; (2)), which occurred ~10 kyrs after the TII rise in DH $\delta^{18}\text{O}$ (6). The extended DH2 $\delta^{18}\text{O}$ record shows similar agreement in the timing of a multi-metre drop in the palaeo water table (e.g., local drying) associated with TIII-VI (27, 28, 34) (Fig. 2).

- Line 1376: The DH2 $\delta^{13}\text{C}$ timeseries is inversely related to DH2 $\delta^{18}\text{O}$ (Fig. 2), such that enriched DH2 $\delta^{13}\text{C}$ broadly corresponds to periods of depleted DH2 $\delta^{18}\text{O}$ and high palaeo water tables (27, 28).
- Line 1569: Reversals in DH2 $\delta^{13}\text{C}$ coincide with a rapid lowering of the DH2 water table from glacial high stands (+9-10 m) to levels similar to today (27, 28). The DH2 $\delta^{13}\text{C}$ reversal associated with MIS 5e, 7e and 9 occur when the palaeo water table reaches below a threshold of +3.7 m (Fig. 2), which is equivalent to +52% recharge relative to today (25). Declining effective moisture coupled with warm temperatures in the first half of interglacial periods triggered a loss of vegetation density in the high-elevation recharge centers of DH2. This loss continued throughout the latter half of each interglacial. In total, the extended DH2 $\delta^{13}\text{C}$ record suggests that seasonality is the dominant driver of orbital-scale environmental change in the highlands of southwest Nevada, with a tipping point in effective moisture (<50% greater recharge relative to today) that result in a rapid and unilateral decline in primary productivity during warm interglacials.

Figure 2: Devils Hole 2 $\delta^{18}\text{O}$, $\delta^{13}\text{C}$, and paleo water table elevation over the last 600 ka. From top: DH2 $\delta^{18}\text{O}$ and $\delta^{13}\text{C}$ (10; this study). Orange lines indicate core-P $\delta^{18}\text{O}$ data spliced into core D record (see text). Blue diamonds indicate paleo water table elevations in DH2 cave relative to modern, including 2σ age uncertainties (25, 26). Blue shading is produced from a binomial spline of DH2 water table data. Dashed vertical lines indicate prominent reversals in the DH2 $\delta^{13}\text{C}$ record, which coincide with water table decline.

Oxygen isotope results and interpretation. I really like using the isotope-enabled GCM to support this work – it provides a compelling explanation of the circulation changes from glacial to interglacial climates and strengthens the paper overall – especially the finding of source regions in the Pacific changing. However, as I note below, digging into the model results in more detail would help explain why the glacial isotopic values are more negative – and it would help explain why the amplitude of isotopic change in the model is significantly less than in the observations (even as it is in the right direction).

We agree that the original manuscript did not sufficiently explain GCM results (owing, in part, to word count). We have since significantly expanded this section starting on line 531:

- To further investigate potential mechanisms, we used a water isotope-enabled Earth System Model (iCESM1.3) with moisture tagging to examine changes in $\delta^{18}\text{O}$ of precipitation at DH2 (52, 53). LGM simulations show a 1.3‰ decrease in the annual average $\delta^{18}\text{O}$ of precipitation at DH2 relative to preindustrial (PI; 1850 CE) (Fig. 3). A depletion of water vapour $\delta^{18}\text{O}$ for all months is partially attributed to cooler LGM temperatures at DH2's moisture source regions. LGM simulations show a 50% increase in annual precipitation amount (Fig. 3).

Increased precipitation occurred during winter months (Fig. 3; Fig. S4) during which there was enhanced transport of Pacific-sourced moisture. This is attributed to a southerly displaced Pacific Storm Track, as supported by the LGM-PI differences in the mean eddy kinetic energy (Fig. S6b) and an increase in the vapour fraction sourced from the southern North Pacific (10°N to 30°N) relative to northern North Pacific (30°N to 60°N) (Fig. S4). Despite its lower latitude source, southern North Pacific water vapour arrives to DH2 ~2‰ more depleted relative to northern North Pacific vapour (Fig. 3) likely due to longer moisture trajectory paths and/or higher rainout efficiency resulting from a greater land-sea temperature gradient. The proportion of precipitation sourced from the North American continent decreased during the LGM (Fig. 3) due to suppressed re-evaporation from land sources as a result of cooler terrestrial surface temperatures. This iCESM insight agrees with recent a DH2 17Oexcess study that suggest reduced continental recycling during glacials (54). Because land-sourced vapour is relatively enriched in $\delta^{18}\text{O}$, a decrease in its contribution to DH2 precipitation during the LGM results in an overall depletion (Fig. 3). Finally, model results do not support a correlation between $\delta^{18}\text{O}$ change and precipitation amount. This finding agrees with proxy data, which show a decoupling of DH2 $\delta^{18}\text{O}$ from local effective moisture at various points in time. For example, MIS 5e DH2 $\delta^{18}\text{O}$ reaches maximum interglacial values at approximately 127 ka before plateauing for ~6 kyrs (3, 12, 13), whereas the DH2 water table continues dropping until 120.3 \pm 0.5 ka. A similar decoupling of DH2 $\delta^{18}\text{O}$ and local effective moisture is observed during interglacials MIS 7e, 7c, 9, and 11, within dating uncertainties.

In total, iCESM moisture tagging experiments suggest two key drivers of DH2 $\delta^{18}\text{O}$ variability on glacial-interglacial timescales. First, vapour delivered to DH2 during the LGM is more strongly depleted in $\delta^{18}\text{O}$ for all months due to cooler temperatures and temperate-driven rainout effects (Fig. S4). Second, a change in the proportion of moisture from distinct sources, specifically (i) an increase in depleted moisture from the southern North Pacific due to a southward displacement of the Pacific Storm Track and (ii) a decrease in moisture from the North American continent due to decreased continental recycling during the LGM, as corroborated by proxy data (54). iCESM does not fully resolve the NAM; we therefore cannot rule out NAM-related processes as potential contributors to DH2 $\delta^{18}\text{O}$ on glacial-interglacial timescales. Simulated change in $\delta^{18}\text{O}$ between LGM and PI ($\Delta\delta^{18}\text{O}$) underestimates the observed $\Delta\delta^{18}\text{O}$ in DH2 record by ~1‰ (considering seawater corrections). This may be due to (1) limitations in iCESM to simulate processes related to NAM strength, (2) lower LGM-PI temperature differences in iCESM simulations ($\Delta 5^\circ\text{C}$) relative to the truly magnitude suggested by proxy reconstructions, and/or (3) inaccuracies in iCESM ice volume forcing in the southern Sierra Nevada mountain range (62, 63), which may alter moisture trajectories and contribute to increased rainout during glacial periods.

The statistical work on this well-dated record is outstanding. The phase lags to various forcing parameters including precession and obliquity, sea level, and atmospheric carbon dioxide provide many insights into the nature of climate forcing and response and is one of the strongest aspects of the paper.

Finally, the paper ends abruptly without a strong concluding paragraph (perhaps that is the journal format) but I didn't think the paper fully brought out what the links between Great Basin hydroclimate and forcing tells us about implications for future warming and drying. More detail here would strengthen the manuscript.

We agree and have added a concluding paragraph, starting on line1637:

- In summary, the phasing of DH2 $\delta^{18}\text{O}$ timeseries suggests that temperature-related processes are dominant drivers of orbital scale $\delta^{18}\text{O}$ variability in southern Great Basin precipitation, consistent with iCESM model outputs. Global ice volume lags DH2 $\delta^{18}\text{O}$ on average, suggesting that mechanisms linked to the North American ice sheets (e.g., shifting storm track) contributed to, but were not the primary driver of, DH2 $\delta^{18}\text{O}$ variability. In contrast, the DH2 $\delta^{13}\text{C}$ timeseries is in-phase with seasonal insolation during the warm growing season (between summer solstice and fall equinox). Prominent $\delta^{13}\text{C}$ lows, indicating

high primary productivity, coincide with peak boreal summer insolation during the last six interglacial periods. A rapid decrease in vegetation density coincides with warm interglacial temperatures and <50% greater recharge relative to today, indicating a past tipping point for environmental decline. This study sheds new light on the relationship between temperature, moisture balance, and vegetation in the southern Great Basin on orbital timescales. Our results underscore the link between increased CO₂ concentrations and regional warming (17, 18, 71, 72), which is expected to contribute to reduced effective moisture in the Great Basin over the coming century (15, 19).

The figures are clear and support the manuscript well. Same for the supplementary material.

Specific Comments

Abstract:

Lines 44-45: It is a 736 kyr record, but abstract should acknowledge the ~80 kyr gap from 580 to 660 ka. This becomes important in the discussion where the most robust statistical analysis is really for the younger part of the record.

See response below.

Lines 48-49: States that the statistical analysis was for the whole record but in the discussion, the analysis appears to pertain to the continuous record from 0 ka to 580 ka. The earliest part of the record does not appear to be part of the analysis. (Lines 244-245 make clear that phase relationships are calculated only over the last 500 kyr.)

Abstract has been changed to the following:

- The Great Basin in the southwest United States experienced major hydroclimate shifts throughout the Quaternary. Understanding the drivers behind these past changes has become increasingly important in improving future climate projections. Here, we present an absolute-dated $\delta^{18}\text{O}$ and $\delta^{13}\text{C}$ record from Devils Hole cave 2 (southern Nevada) that reveals climate and environmental changes in the southern Great Basin over the last 580,000 years. Water isotope-enabled Earth system simulations and phasing analysis show that temperature-related mechanisms are a primary driver of $\delta^{18}\text{O}$ variability, with additional drivers stemming from processes linked to North American ice. Vegetation density in the highlands of southwest Nevada is forced by Northern Hemisphere summer seasonality. A rapid decline in primary productivity occurs during warm interglacial periods when local groundwater recharge reaches <50% greater than today. Our study sheds new light on the relationship between temperature, moisture balance, and vegetation over the last six glacial-interglacial cycles.

To increase clarity, we also adjusted the title to: Controls on the southwest USA hydroclimate over the last six glacial-interglacial cycles

Line 56: The implications for future warming and drying are not specified here nor are they really discussed at all in the text. Either remove this sentence, or add some more discussion on the implications of the record for future warming and drying (preferred). This should also be added at the end of the paper.

Agreed. This line has been adjusted to the following:

- Understanding these mechanisms has become increasingly urgent, as warmer temperatures over the next century are expected to reduce water availability in this already water-scarce yet increasingly populated region (15-19).

And two sentences in the concluding paragraph were added:

- This study sheds new light on the relationship between temperature, moisture balance, and vegetation in the southern Great Basin on orbital timescales. Our results underscore the link between increased CO₂ concentrations and regional warming (17, 18, 71, 72), which is expected to contribute to reduced effective moisture in the Great Basin over the coming century (15, 19).

Body of Manuscript

Paragraph beginning on line 140 and Figure 1: The calcite $\delta^{18}\text{O}$ results show an amplitude of about 2 to 2.5 per mil from glacial maxima to interglacials, whereas the modeling results presented here show amplitudes of no more than 0.6 per mil, and the favored pathway from SNP is closer to 0.5 per mil. These are in the right direction (i.e. more negative for glacials) but these differences in amplitude should be explicitly acknowledged in the discussion. The paragraph gives some explanations as to why the glacials are isotopically lighter citing earlier work – do your model results bear these out? More could be diagnosed from the model to strengthen these interpretations, and it would be helpful to understand why the model underestimates the degree of change.

Addressing these questions, we have added the following text starting on line 613:

- Simulated change in $\delta^{18}\text{O}$ between LGM and PI ($\Delta\delta^{18}\text{O}$) underestimates the observed $\Delta\delta^{18}\text{O}$ in DH2 record by $\sim 1\%$ (considering seawater corrections). This may be due to (1) limitations in iCESM to simulate processes related to NAM strength, (2) lower LGM-PI temperature differences in iCESM simulations ($\Delta 5^\circ\text{C}$) relative to the truly magnitude suggested by proxy reconstructions, and/or (3) inaccuracies in iCESM ice volume forcing in the southern Sierra Nevada mountain range (62, 63), which may alter moisture trajectories and contribute to increased rainout during glacial periods.

The moisture tagging experiments shown in Figure S5 show the differences in isotopic composition of precipitation from different source regions, but can they also show the effects of enhanced rainout efficiency and cooler STs?

This was originally argued in Tabor et al. 2021 which also looked at iCESM LGM time slices. To clarify this point, we added the following to line 525:

- Tabor et al. (2021) suggest that a greater land-sea temperature gradient during the LGM increased the rainout efficiency of moisture trajectories moving inland to the Great Basin. This effect, when coupled with cooler LGM temperatures that suppressed evaporation, would result in lower $\delta^{18}\text{O}$ of precipitation during glacial periods (40, 50).

Lines 181- 182:

Lines 317-326: Discussion of controls of $\delta^{18}\text{O}$ includes several processes including temperature dependent rainout of moisture, pathway of storms etc. but doesn't fully draw on the isotope enabled modeling presented earlier in the manuscript. It would strengthen the discussion to use more of the modeling results.

Agreed. This paragraph has been changed to the following (starting on line 1366):

- Overall, our phasing and midpoint analyses suggest that DH2 $\delta^{18}\text{O}$ bears the closest structural and temporal similarity to atmospheric CO_2 variability, both when calculating an average over the entire record and over short periods of abrupt warming and drying. In the event of a termination, rising CO_2 concentrations would warm eastern North Pacific and western North America regions which, in alignment with our iCESM results, would (1) enrich the $\delta^{18}\text{O}$ of water vapor from moisture sources, (2) decrease land-sea temperature gradients (decreasing rainout efficiency along moisture trajectories), and (3) increase evaporation from the North American continent, collectively resulting in an enrichment of $\delta^{18}\text{O}$ moisture arriving to DH2. Phasing using the lagged correlation method and midpoint analysis shows that, on average, DH2 $\delta^{18}\text{O}$ leads changes in global ice volume. This suggests that the modulation of the Pacific Storm Track by North American ice sheets may have contributed to (e.g., amplified), but was not a primary driver of, changes in DH2 $\delta^{18}\text{O}$ on orbital timescales. Another example of the close coupling between DH2 $\delta^{18}\text{O}$ and CO_2 is at the onset of terminations. The timing of DH2 $\delta^{18}\text{O}$ rise closely aligns with the onset of CO_2 rise associated with TII-V, including the “early” rise in DH2 $\delta^{18}\text{O}$ and CO_2 at the onset of Termination IV (358-345 ka) which is absent from global ice volume records (Fig. 5).

Line 319: Great Basin $\delta^{18}\text{O}$ variability – clarify if you mean the $\delta^{18}\text{O}$ of precipitation or of the calcite. Adjusted.

Line 322: Storm systems move across western North America
Corrected.

Lines 343-344: This section promises insights into periods of rapid warming and drying of the southern Great Basin associated with the last 8 glacial terminations, but only goes into detail for the youngest terminations and has much less detail for TIII to TV, and no discussion of the earlier glacial terminations (one of which occurs during the hiatus so can't be discussed). Either include more discussion of the earlier terminations (preferred) or modify the 8 glacial termination wording.

The following has been added starting on line 1330:

Lastly, we investigated potential drivers of DH2 $\delta^{18}\text{O}$ during periods of abrupt warming and drying. To do so, we determined the midpoint of TI-V in DH2 $\delta^{18}\text{O}$ and the aforementioned records on their individual chronologies. The midpoint of the ascending limb of DH2 $\delta^{18}\text{O}$ during TII (132.15 ± 1.5 ka) was previously calculated by (10). Using the same approach, we calculate the midpoint associated with TIII (244.0 ± 1.1 ka), TIV (341 ± 3 ka), TV (430 ± 6 ka), TVI (529 ± 6 ka) and TVIII (708 ± 8 ka) of the DH2 $\delta^{18}\text{O}$ record (table S5; see methods). Maximum groundwater residence time (880 yrs) falls within chronological uncertainties of each midpoint. As shown in (10), multiple growth rate changes in core D during TI and the Holocene resulted in poor age control (up to 18% relative 2σ uncertainty between 18-10ka). For this study, we focus on TII-TV during which the DH2 record has the best age control ($\leq 1\%$ relative 2σ uncertainties) (Fig. 6).

Lines 343-344: Does age uncertainty preclude discussion of the earlier pre-mid Bruhnes terminations?
Correct, as explained in the above response.

Lines 405-406: Refers to Wend et al. 2018 but not the Steidle et al. 2024 paper on water table variations. Seems appropriate to refer to the recently published DH paper.
Please refer to our first response to reviewer 3.

Lines 409-411: Sentence states that the record provides new constraints on relative P-ET and temperature thresholds, but those are not explicitly stated here. Temperature thresholds for deforestation tipping points would be of broad interest but it is unclear what they are and for that matter, how close we are now to such a tipping point. More detail is needed here.

The following sentences were added to emphasize this point (without over-emphasizing, as pointed out by reviewer 1):

- Line 1699: Reversals in DH2 $\delta^{13}\text{C}$ coincide with a rapid lowering of the DH2 water table from glacial high stands (+9-10 m) to levels similar to today (27, 28). The DH2 $\delta^{13}\text{C}$ reversal associated with MIS 5e, 7e and 9 occur when the palaeo water table reaches below a threshold of +3.7 m (Fig. 2), which is equivalent to +52% recharge relative today (25). Declining effective moisture coupled with warm temperatures in the first half of interglacial periods triggered a loss of vegetation density in the high-elevation recharge centers of DH2. This loss continued throughout the latter half of each interglacial. In total, the extended DH2 $\delta^{13}\text{C}$ record suggests that seasonality is the dominant driver of orbital-scale environmental change in the highlands of southwest Nevada, with a tipping point in effective moisture (<50% greater recharge relative to today) that result in a rapid and unilateral decline in primary productivity during warm interglacials.

The paper ends somewhat abruptly without strong concluding remarks about the main findings.
Agreed, we have now added a concluding paragraph.

Supplemental material:

Figure S15 shows $d^{13}\text{C}$ plotted vs DH and DH2 paleo water table elevations from 1994 and 2018 papers, but not the 2024 paper just published a month ago. Why not? This figure goes back only 350,000 years, but it could be taken all the way back to ~750,000 years in comparison with the Steidle et al. 2024 paper. (This comment goes to the lack of citation of any of the work from the 2024 water table paper).

Please refer to our first response to reviewer 3.

REVIEWERS' COMMENTS

Reviewer #1 (Remarks to the Author):

Re-review of Controls on the southwest USA hydroclimate over the last seven glacial-interglacial cycles for Nature Communications

I have re-read the paper and the response document from the authors. The authors have done an extremely thorough job in responding to, addressing and expanding upon the ideas/comments from me and the other reviewers. I have no further comments, this works should be published and I anticipate that this will be a highly cited and discussed record for decades. The precise interpretation of the record may change but the secure chronology and robust dataset are essentially unprecedented in terrestrial paleoclimatology.

Reviewer #2 (Remarks to the Author):

I have reviewed the revised version of Wendt et al. and found that the authors have addressed all of my previous concerns. The manuscript has improved considerably and is now close to publication. I only have a few very minor suggestions, none of which should prevent acceptance:

Figure 1: Please consider including a map showing the study area. While I am familiar with the site, many readers, particularly those outside the United States, may not know where it is located.

Done

Line 105: Remove the phrase “the using.”

Done

Line 150: Change “...due an early warming...” to “...due to early warming....”

Done

Consistency: The manuscript uses both “paleo” and “palaeo.” Please standardize to one form throughout.

Done

Line 172: Revise “...exceptionally wet...” to “...was exceptionally wet....”

Done

Line 238: Revise “...with recent a DH2...” to “...with a recent DH2....”

Done

Line 301: Revise “...linked to mechanisms may influence...” to “...linked to mechanisms that may influence....”

Done

Reviewer #3 (Remarks to the Author):

Review of “Controls on the southwest USA hydroclimate over the last six glacial-interglacial cycles”

Wendt et al.

Resubmitted to Nature Communications

I reviewed an earlier draft of this manuscript and had several suggestions on oxygen isotopes results and interpretation (and using more of the isotope enabled GCM results), using dates and data from the Wendt et al 2024 paper), and the inclusion of a strong concluding paragraph.

After reading through the revised manuscript and the authors’ comments to reviewers, I am very pleased with the resulting manuscript. The authors have done a great job addressing the comments of all three

reviewers and I think the paper is now acceptable for publication (pending a few very minor comments below).

The revised oxygen isotope change discussion is really strong and the authors then use this to tie isotopes into the statistical analysis to more strongly support interpretations of the climate change mechanisms, especially at glacial terminations. While this apparent before, I feel that the increased discussion of isotopic control mechanisms makes the climatic change interpretations stronger. I also feel that the key points here are improved by removing the millennial-scale mechanisms (per reviewer 2) and focusing on the glacial mid-point analysis.

The new Figure 2 with the DH2 water table data works well and shows the power of combining these proxies and enhances the overall story. The added concluding paragraph does a nice job of bringing the main points of the manuscript home and the paper is greatly improved by adding this.

I don't have any other substantive comments and I really like the way this manuscript has evolved. It is ready for publication with a few minor points below:

Abstract: line 63 – what do you mean by N.H. summer seasonality – does this mean intensity?
Yes, abstract now reads “Northern Hemisphere summer intensity”

Line 65 – “when local recharge reaches < 50% greater than today” is confusing. Would it be better to say “when local recharge declines to <50% of modern”? or something to that effect?
Adjusted to “A rapid decline in primary productivity occurs during warm interglacial periods when local groundwater recharge declines to <50% above modern.”

References: Line 1598 or reference 38. Incomplete – if this is a MS thesis, should it have the full title etc.? (I didn't catch this before).
This was an EndNote mistake. All references have been checked and corrected.

Reviewer 1 suggested some additional references wrt Great Basin lakes that record multiple glacial-interglacial cycles – there is also a new long sediment core record from Stoneman Lake in central AZ (not part of the Great Basin, but relatively close to DH) that records multiple wet glacial - dry interglacial lake alternations. (Staley et al., 2022 GSA Bulletin)
Added to line 67. We also removed “Great Basin” from the sentence to include records from the greater southwest USA.